# Physiochemical interaction between osmotic stress and a bacterial exometabolite promotes plant disease

Felix Getzke[1,7], Lei Wang[2,7], Guillaume Chesneau[1], Nils Böhringer[2,3], Fantin Mesny[1,6], Nienke Denissen[1], Hidde Wesseler[1], Priscilla Tijesuni Adisa[1], Michael Marner[4], Paul Schulze-Lefert[1,5], Till F. Schäberle[2,3,4] ✉ & Stéphane Hacquard[1,5] ✉

Various microbes isolated from healthy plants are detrimental under laboratory conditions, indicating the existence of molecular mechanisms preventing disease in nature. Here, we demonstrated that application of sodium chloride (NaCl) in natural and gnotobiotic soil systems is sufficient to induce plant disease caused by an otherwise non-pathogenic root-derived *Pseudomonas brassicacearum* isolate (R401). Disease caused by combinatorial treatment of NaCl and R401 triggered extensive, root-specific transcriptional reprogramming that did not involve down-regulation of host innate immune genes, nor dampening of ROS-mediated immunity. Instead, we identified and structurally characterized the R401 lipopeptide brassicapeptin A as necessary and sufficient to promote disease on salt-treated plants. Brassicapeptin A production is salt-inducible, promotes root colonization and transitions R401 from being beneficial to being detrimental on salt-treated plants by disturbing host ion homeostasis, thereby bolstering susceptibility to osmolytes. We conclude that the interaction between a global change stressor and a single exometabolite from a member of the root microbiome promotes plant disease in complex soil systems.

Subterranean and aerial plant tissues are colonized by complex microbial communities that are referred to as the root and shoot microbiota, respectively[1–4]. Recent efforts to systematically isolate and characterise the microbiota of healthy-looking *Arabidopsis thaliana* grown in natural soils revealed that few bacteria (<5%) can cause disease under specific laboratory conditions[5–7]. These include for example *Pseudomonas brassicacearum* Root401 (referred to as R401), *Streptomyces spp.* Root107 (R107), *Xanthomonas spp.* Leaf131 and

Leaf148 (L131, L148), or multiple *Pseudomonas viridiflava* isolates that were retrieved from symptomless plant tissues[2,5,8–10]. The identification of these so-called opportunistic pathogens suggests the existence of molecular mechanisms that suppress detrimental phenotypes in nature whilst facilitating infection under laboratory conditions.

Stevens (1960)[11] advanced the concept of the 'disease triangle', by which environmental factors contribute to the establishment of plant diseases. In this triangle, abiotic conditions can influence the

[1]Department of Plant Microbe Interactions, Max Planck Institute for Plant Breeding Research, 50829 Cologne, Germany. [2]Institute for Insect Biotechnology, Justus-Liebig-University Giessen, 35392 Giessen, Germany. [3]German Center for Infection Research (DZIF), Partner Site Giessen-Marburg-Langen, 35392 Giessen, Germany. [4]Fraunhofer Institute for Molecular Biology and Applied Ecology (IME), Branch for Bioresources, 35392 Giessen, Germany. [5]Cluster of Excellence on Plant Sciences (CEPLAS), Max Planck Institute for Plant Breeding Research, 50829 Cologne, Germany. [6]Present address: Institute for Plant Sciences, University of Cologne, 50674 Cologne, Germany. [7]These authors contributed equally: Felix Getzke, Lei Wang. ✉e-mail: Till.F.Schaeberle@agrar.uni-giessen.de; hacquard@mpipz.mpg.de

host, the microbiota, or the interaction between the two, facilitating or inhibiting pathogen progression[12-14]. Similar to pathogen perception, e.g., the sensing of salt stress results in signalling cascades involving cytoplasmic $Ca^{2+}$ influx, ROS production and the accumulation of plant hormones, primarily abscisic acid[15-18] (ABA). ABA mediates closure of stomata to restrict transpirational water loss[18] and inhibits the expression of the salicylic acid (SA) biosynthetic gene *ISOCHORISMATE SYNTHASE1* (*ICS1/SID2*) in *A. thaliana*, thereby suppressing SA-dependent immunity[19,20]. This ABA-SA cross-talk was recently shown to be leaf-age dependent and controlled by the SNAC-A transcription factor cascade[21]. Collectively, abiotic stress signalling and host innate immunity processes are tightly connected and likely contribute to disease emergence in nature[14,22,23].

Here, we focus on a dominant member of the bacterial root microbiota called *P. brassicacearum* R401 that was previously shown to be detrimental in mono-association experiments with *A. thaliana* in an agar matrix-based gnotobiotic system[6]. This strain was also recently shown to deploy unrelated inhibitory exometabolites that co-function to keep bacterial competitors at bay and promote strain colonization success in roots[24]. We report here that R401 is non-pathogenic on plants grown in natural or gnotobiotic peat-based soil systems and that NaCl treatment promotes R401 disease symptoms in these soil-grown plants, providing evidence for environmental conditions that conditionally promote plant disease. In R401, we identify a biosynthetic gene cluster (BGC) homologous to the *syp-syr* BGC that is responsible for syringopeptin biosynthesis in *P. syringae* B728a[25] and we demonstrate that this locus confers root colonization capability to the strain and is sufficient to transition R401 from being beneficial to being detrimental on salt-treated plants. We further report that the produced exometabolite is necessary and sufficient to promote susceptibility to salt stress, thereby bolstering disease (here referred to as the combined detrimental effect of a biotic and an abiotic factor). We conclude that the physiochemical interaction between a bacterial exometabolite and NaCl promotes host susceptibility to osmotic stress and alters host health in complex soil-microbiome systems.

## Results

### R401 is detrimental on soil-grown plants facing salt stress

Inoculation of R401 on *A. thaliana* plants grown on ½ Murashige-Skoog (½ MS) agar matrix plates was shown to cause extensive plant growth inhibition and anthocyanin accumulation in shoots[6] and we were able to reproduce similar phenotypes (Fig. 1a). However, in the sterile peat matrix of the gnotobiotic Flowpot system[26] or in natural, non-sterile Cologne agricultural soil (CAS), R401 inoculation did not cause disease (Fig. 1b,c). Salt stress has been shown to facilitate *Pseudomonas* infections in tomato[27] (*Solanum lycopersicum*, Solanaceae) and cucumber[28] (*Cucumis sativus*, Cucurbitaceae). We therefore speculated that salt stress may facilitate the detrimental effects of *P. brassicacearum* R401 on *A. thaliana* (Brassicaceae). In axenic conditions (Fig. 1b) and in the presence of a complex, natural soil microbiota (Fig. 1c), the application of 100 mM NaCl negatively affected plant growth (HK 0 mM NaCl vs. HK 100 mM NaCl; $p < 0.001$, Kruskal-Wallis followed by Dunn's post-hoc test, Fig. 1b,c). Co-inoculation of R401 and 100 mM NaCl further aggravated this effect in both soil systems, leading to highly stunted and chlorotic plants, reminiscent of R401 effects in ½ MS agar plates (R401 0 mM NaCl vs. R401 100 mM NaCl; $p < 0.001$ and $p < 0.01$ for Fig. 1 b and c, respectively, Fig. 1a-c). Collectively, this indicates that in complex soil systems, the combined action of the opportunistic pathogen R401 and salt stress is required to cause disease on *A. thaliana*.

### Salt-treated plants down-regulate immune processes in shoots but not in roots

We hypothesized that salt stress response in *A. thaliana* comes at the cost of dampening immune sectors in roots, which in turn promotes R401 infection. We sequenced the root and shoot transcriptome of *A. thaliana* 28 days post treatment with either heat-killed (HK) or live R401 wild-type (WT) strains mono-inoculated in the absence or presence of NaCl (0 vs. 100 mM) in the Flowpot system (Fig. 1b, Supplementary Data 1). Inoculation of R401 in the absence of salt stress had marginal effects on both root and shoot transcriptomes compared to control samples (0 mM NaCl + R401 HK), with 2 and 14 differentially

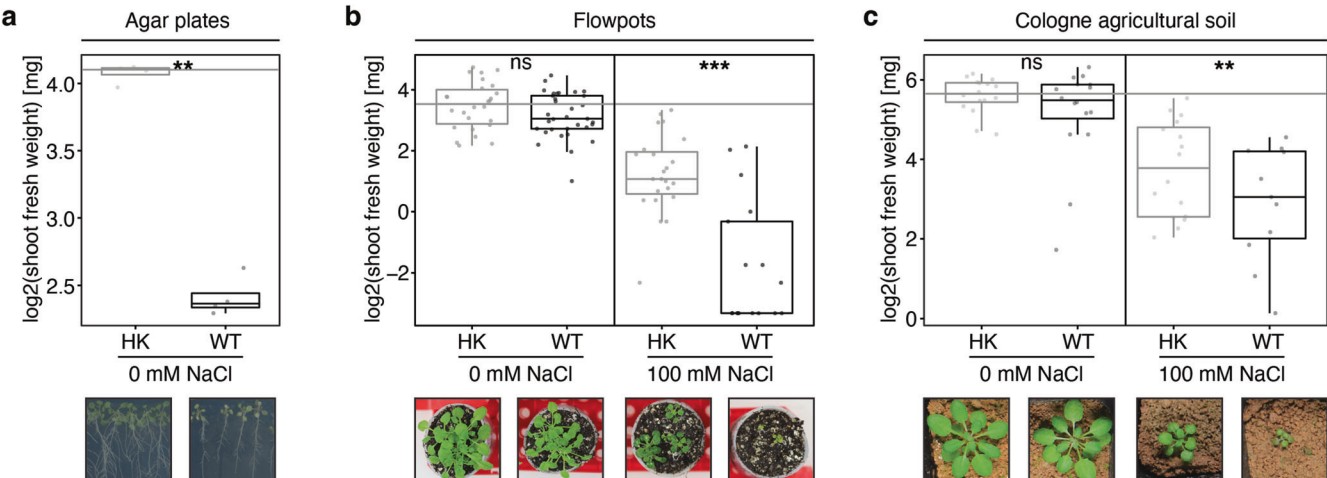

**Fig. 1 | Detrimental activity of *P. brassicacearum* R401 is facilitated by salt stress in soil. a** log2-transformed shoot fresh weight of *A. thaliana* plants grown axenically on ½ MS agar plates for 19 dpi. At 14 dpi plants were flushed with either heat-killed (HK) or live wild-type (WT) R401 cells. Five minutes after flushing plants were transferred to new sterile plates and grown for five days (*n* = 5 plants). **b** log2-transformed shoot fresh weight of *A. thaliana* plants grown in the gnotobiotic Flowpot system for 28 dpi in the presence or absence of 100 mM NaCl and either heat-killed (HK) or live wild-type (WT) R401 cells (*n* = 30 plants). **c** log2-transformed shoot fresh weight of *A. thaliana* plants grown in the non-sterile Cologne agricultural soil (CAS) in the greenhouse for 28 dpi in the presence or absence of 100 mM NaCl and either heat-killed (HK) or live wild-type (WT) R401 (*n* = 16 plants). **a**–**c** Representative images illustrating the respective plant phenotypes are shown below each plot. Within each figure panel, all images are to scale. Statistical significance was determined by Kruskal-Wallis followed by Dunn's post-hoc test and Benjamini-Hochberg adjustment. Significance compared to HK is indicated by black asterisks (∗∗ and ∗∗∗ indicate $p < 0.01$, and 0.001, respectively; ns, not significant). Statistical comparisons were conducted between WT and HK samples for each NaCl treatment separately. Boxplots show 25th percentile, median, and 75th percentile.

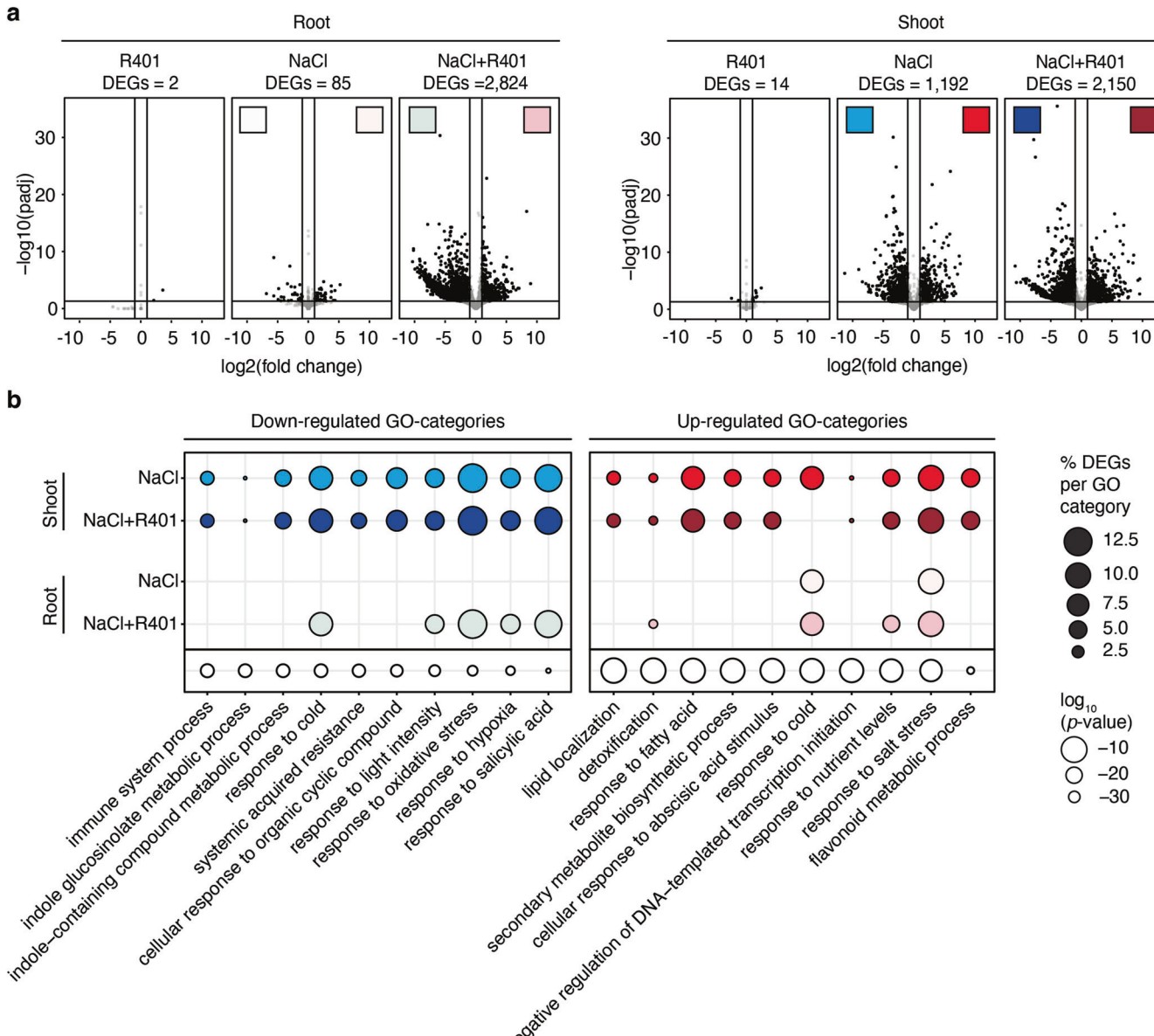

**Fig. 2 | Interaction between R401 and NaCl triggers extensive transcriptional reprogramming in roots. a** Transcriptomic analysis of *A. thaliana* shoots and roots (28 dpi) grown in the gnotobiotic Flowpot system in the presence of R401, 100 mM NaCl or 100 mM NaCl and R401 compared to the control condition (0 mM NaCl with Heat-Killed (HK) R401), *n* = three biological replicates each comprising 10 plants. The number of differentially-enriched genes (DEGs) as compared to the control condition is indicated above the graphs and are shown in black in the volcano plots. **b** Significantly enriched GO-categories in NaCl or NaCl + R401 treated shoots or roots. All significantly up-or down-regulated genes (*p.adj* ≤ 0.05 and fold change ≤ 0.5 or ≥ 2) as compared to control conditions (0 mM NaCl + HK R401) were selected for the analysis. Cluster-based GO-category enrichment analysis was performed using Metascape[36] in multiple gene lists-mode for significantly down- and upregulated genes separately. Spheres size indicates the number of genes (in %) that are significantly up-regulated for each GO-category. The absence of a sphere indicates that the respective GO-category was not significantly enriched in the respective condition. *p*-values were calculated based on the cumulative hypergeometric distribution and are indicated in unfilled spheres.

expressed genes identified (DEGs), respectively ($p < 0.05$, −1 <Log2FC > 1, Fig. 2a). Salt stress alone also triggered subtle transcriptional reprogramming in roots (85 DEGs, Fig. 2a and Supplementary Fig. 1a), but not in shoots (1,192 DEGs, Fig. 2a and Supplementary Fig. 1b). Based on PANTHER functional annotation analyses[29–31], up-regulated genes in roots of salt-treated plants can be categorized into the GO-terms 'response to abscisic acid', 'response to osmotic stress', and 'response to water deprivation', while in shoots the most enriched GO-terms included 'anthocyanin-containing compound biosynthetic process', 'hyperosmotic salinity response' and 'abscisic acid-activated signaling pathway'. Collectively, these data confirmed that salt stress as applied in our experimental setup activated the stereotypical salt

stress response and interfered with plant photosynthesis[32–35]. Notably, combinatorial treatment of R401 and salt had the most extensive effect on the host transcriptome, with 2,824 and 2,150 DEGs identified in roots and shoots, respectively (Fig. 2a and Supplementary Fig. 1a,b). However, only in roots did R401 cause major additive transcriptional reprogramming compared to salt stress alone, with 1,922 newly identified DEGs (Fig. 2a). Although this activation was associated with disease symptoms, no increased proliferation of R401 was noted in roots of salt-treated compared to control plants, indicated that bacterial over-proliferation at roots is likely not the direct cause driving disease (Supplementary Fig. 1c). Cluster-based GO-category enrichment analysis[36] of DEGs responding to salt stress or combinatorial treatment

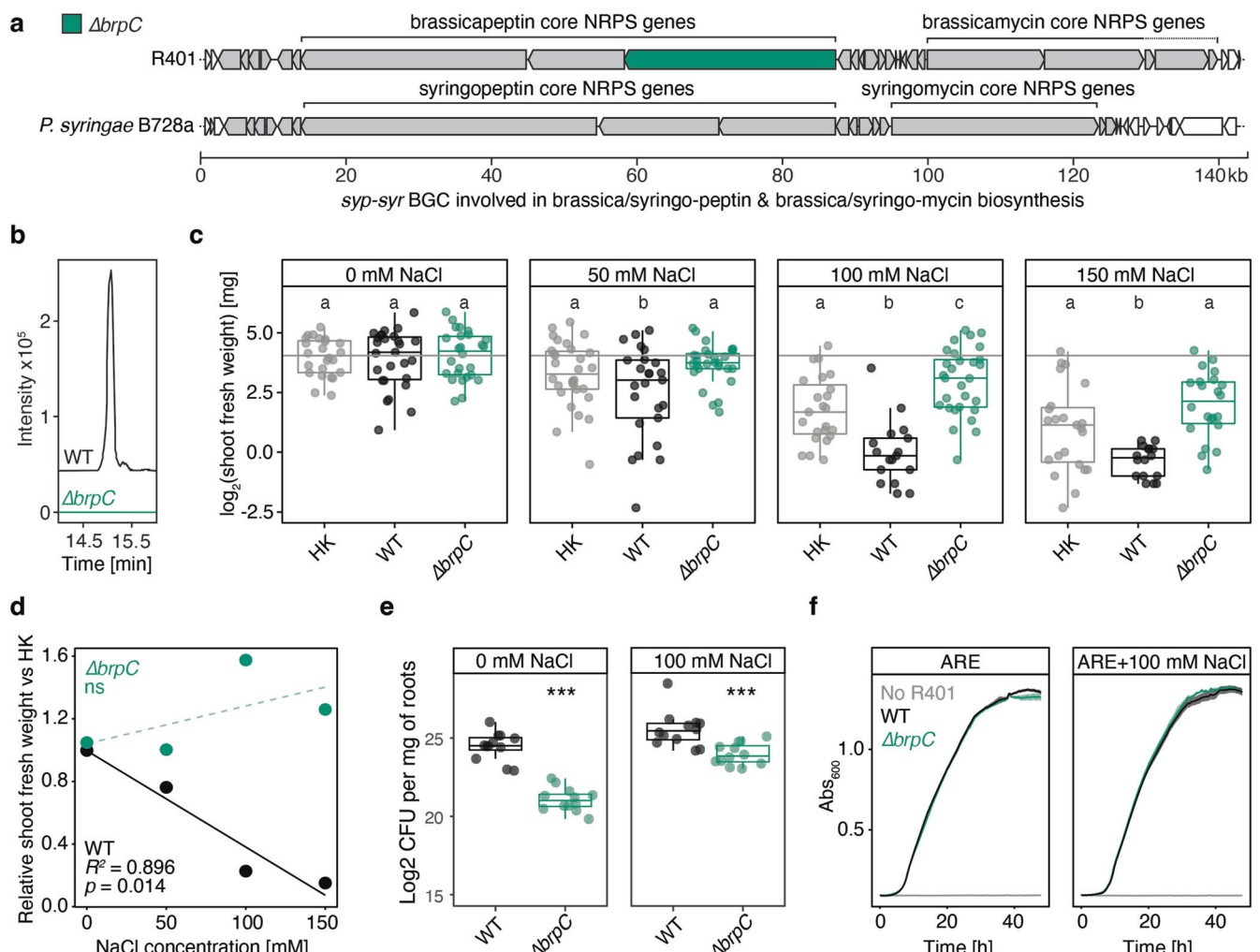

**Fig. 3 | Brassicapeptin production is required for R401 detrimental activity on salt-treated plants. a** Schematic overview of the genomic context of the fragmented *syp-syr* operon which encodes genes for syringopeptin and syringomycin biosynthesis in *Pseudomonas syringae* B728a and likely for related specialized metabolites in *Pseudomonas brassicacearum* R401, thereby termed brassicapeptin and brassicamycin. Genes within the biosynthetic gene cluster (BGC) are colored in grey, R401 *brpC* is highlighted in green. *brpC* is likely involved in brassicapeptin biosynthesis in R401. A *ΔbrpC* knockout mutant has been generated in R401, lacking the full-length *brpC* coding region. Gene prediction and annotation was performed using antiSMASH[40] for B728a and R401 separately, the figure was assembled in Adobe Illustrator. **b** Extracted ion chromatograms for R401 brassicapeptin (EICs: $C_{96}H_{161}N_{23}O_{26}$ [M + (1-3)H]$^{(1-3)+}$ ± 0.01 *m/z*) of the WT and mutant extracts, confirming complete lack of brassicapeptin production in the tested mutant. **c** log2-transformed shoot fresh weight of *A. thaliana* plants grown in the gnotobiotic Flowpot system for 28 dpi in the presence of increasing concentrations of NaCl (0, 50, 100, or 150 mM NaCl) and either heat-killed (HK), live wild-type (WT) or live *ΔbrpC* R401 cells. Letters indicate statistically significant differences as

determined by Kruskal-Wallis followed by Dunn's post hoc test and Benjamini-Hochberg adjustment with $p < 0.05$ ($n = 30$ plants). Statistical comparisons were conducted for each salt treatment separately. **d** Correlation analysis of the mean effect of either R401 WT or *ΔbrpC* on *A. thaliana* shoot fresh weight, normalized by the respective HK control and the applied salt concentrations. Data derives from (**b**). *p*-values and $R^2$ derive from a linear regression; ns, not significant. **e** Log2-transformed Colony Forming Units (CFUs) of R401 WT and *ΔbrpC* R401 per gram of *Arabidopsis thaliana* roots. Plants were grown in a gnotobiotic Flowpot system for 28 days in two concentrations of NaCl (0 and 100 mM NaCl). Stars (***) indicate statistically significant differences between WT and *ΔbrpC* ($n = 12$ biologically independent samples each comprising 5 roots), as determined by Student's t-test with $p < 0.001$. Statistical comparisons were conducted for each salt treatment separately. **f** Growth curves of R401 WT and *ΔbrpC* mutant in artificial root exudate (ARE) liquid medium or ARE medium supplemented with 100 mM NaCl; $n = 10$ biologically-independent samples. **c, e** Boxplots show 25th percentile, median, and 75th percentile.

---

of salt and R401 revealed 14 GO-term clusters involved in primary metabolism or biotic and abiotic stresses (Supplementary Fig. 1d). Top 10 GO-terms that showed the strongest down-regulation are dominated by innate immune processes such as 'response to SA', 'innate immune process', 'systemic acquired resistance', or 'indole glucosinolate metabolic processes' (Fig. 2b). However, these terms were specifically down-regulated in shoots but not in roots in response to salt, indicating that salt-induced dampening of immune response was shoot-specific.

## Mutation of host NADPH oxidase *RBOHD* does not promote R401 detrimental activity

We next tested whether salt stress-mediated suppression of immunity in shoots could promote pathogenicity of other opportunistic pathogens, such as *Streptomyces sp.* R107 and *Xanthomonas sp.* L131[6,7]. Resistance to *Xanthomonas* L131, but also another closely related L148 opportunistic pathogen was recently shown to fully depend on the host NADPH/respiratory burst oxidase homologue D[7,9,10] (RBOHD), which was approx. 51-fold down-regulated by salt stress in *A. thaliana*

shoots in our dataset ($p.adj < 0.05$, Supplementary Data 1). Therefore, we hypothesized that salt stress might bolster infection of opportunistic pathogens by dampening ROS-mediated immunity in *A. thaliana*. However, while R401 and R107 became more detrimental on salt-treated *A. thaliana* than on control plants, this was not the case for L131 (Supplementary Fig. 2a). In contrast—and consistent with previous work[7]—inoculation of L131 on an immunocompromised *A. thaliana rbohD* mutant promoted disease ($p < 0.05$). However, this infection facilitation phenotype was not observed for R401, indicating that R401 detrimental activity did not require impairment of RBOHD-dependent ROS production in *A. thaliana* (Supplementary Fig. 2b). Taken together, our results suggest that disease caused by combinatorial treatment of R401 and salt is not associated with bacterial overgrowth in roots, nor with downregulation of root immunity or dampening of RBOHD-dependent ROS-mediated protective immunity.

### R401 *brpC* is required for detrimental activity in salt-treated plants

Inspection of the genome of R401 revealed the absence of genes encoding the type-III secretion system[37–39], indicating that intracellular delivery of bacterial effectors via this machinery is likely not the cause of R401 detrimental activity in salt-treated plants. Therefore, we hypothesized that the detrimental activity of R401 on salt-treated plants is mediated by the production of specialized metabolites. AntiSMASH-based prediction[40] of BGCs revealed a > 140 kb BGC with high similarity to the *syp-syr* BGC of *Pseudomonas syringae* B728a involved in syringopeptin and syringomycin biosynthesis[25] (Fig. 3a and Supplementary Data 2). The two molecules are commonly co-regulated and required for virulence in *Pseudomonas syringae* strains[41]. R401 has been phylogenetically assigned to the *P. brassicacearum* species[24]; therefore, we termed the putatively produced specialized metabolites 'brassicapeptin' and 'brassicamycin'. To test whether the *syp-syr* BGC contributes to R401 disease symptoms on salt-treated *A. thaliana*, we generated a marker-free knockout mutant (*ΔbrpC*) lacking *brpC*, one of the core biosynthetic genes putatively involved in brassicapeptin biosynthesis (Fig. 3a). Loss of brassicapeptin production in the *ΔbrpC* mutant was further validated by liquid chromatography-mass spectrometry (LC-MS, Fig. 3b). We then mono-inoculated heat-killed (HK) or live R401 wild type (WT) or *ΔbrpC* mutant cells into the gnotobiotic Flowpot system across a gradient of NaCl concentrations (0–150 mM NaCl). R401 WT was again able to promote disease on salt-treated plants, and this effect was NaCl-concentration dependent (Fig. 3c), indicating a dose-dependent relationship between increased salt concentrations and R401 detrimental activity (Fig. 3d). Remarkably, the detrimental effect of R401 WT on salt-treated plants was fully abolished in the *ΔbrpC* mutant, which was even able to partly rescue salt stress-induced plant growth inhibition (Fig. 3c). Therefore, mutation of a single bacterial gene involved in the production of a specialized exometabolite was sufficient to turn this detrimental isolate into a beneficial plant growth-promoting isolate under salt stress. Notably, *brpC*-dependent detrimental activity of R401 under salt stress was retained in plants co-cultured with a representative, yet simplified 15-member synthetic microbial community in the FlowPot system (Supplementary Fig. 3, see methods). In salt-treated plants, *brpC* is a disease determinant that dominantly functions irrespective of the absence or presence of microbial competitors.

Quantification of CFU of R401 WT and *ΔbrpC* in *A. thaliana* roots (0 and 100 mM NaCl) revealed impaired root colonization of the mutant, irrespective of the salt conditions (Fig. 3e). Notably, this growth defect was not observed in axenic liquid medium (Fig. 3f), indicating that *brpC* is a root colonization determinant. Notably, the growth of R401 WT was insensitive to 100 mM NaCl in liquid medium (Fig. 3f) and did not differ in roots of control and NaCl-treated-plants (Fig. 3e), corroborating our earlier observation (Supplementary

Fig. 2c) that disease was not associated with bacterial overgrowth in salt-treated *A. thaliana*. However, in vitro experiments in a liquid medium revealed that the production of brassicapeptin is salt-inducible (Supplementary Fig 4), raising the possibility that disease symptoms under salt are exacerbated by brassicapeptin over-production at roots, rather than bacterial over proliferation.

### R401 *brpC* drives root colonization and promotes disease in salt-treated tomato

We first tested whether detrimental activity of R401 WT also occurred in the context of other abiotic stresses, such as drought or low photosynthetically active radiation (low PAR[42]). While low PAR had more severe effects on shoot fresh weight compared to drought, this treatment did not facilitate R401-induced disease symptoms. In contrast, R401 in conjunction with drought stress—mimicked by the application of 5% polyethylene glycol (PEG8000)—led to disease, demonstrating that the detrimental activity of R401 is potentiated by hyperosmotic stresses ($p < 0.05$, Supplementary Fig. 5). Next, we assessed whether R401 *brpC* also act as a root colonization and disease determinant in other evolutionary distant plant species. Salt- and *brpC*-dependent detrimental activity of R401 was recapitulated in *Solanum lycopersicum cv.* Micro-Tom (Micro-Tom, $p < 0.05$), but not in *Lotus japonicus* Gifu (Gifu) (Supplementary Fig. 6a,b). This is potentially explained by the fact that the latter plant exhibited high tolerance to the applied salt conditions (Supplementary Fig. 6b) and/or is not affected by brassicapeptin production. We next harvested roots and shoots of both Micro-Tom and Gifu seedlings and quantified colonisation capability of R401 WT and *ΔbrpC* mutant in the presence or absence of NaCl. While colonisation of R401 WT remained stable, irrespective of the salt treatment or the host plant, the *ΔbrpC* mutant showed reduced root colonization in Micro-Tom plants ($p < 0.01$), which was not observed in roots of Gifu plants (Supplementary Fig. 6c,d). Unlike for *A. thaliana* (Fig. 3e), defect in root colonization observed for the *ΔbrpC* mutant was more pronounced in roots of salt-treated Micro-Tom than in roots of control Micro-Tom plants, Supplementary Fig. 6c), suggesting that interaction between brassicapeptin and salt is needed to drive R401 colonization in Micro-Tom. Irrespective of this difference, our data suggest a conserved functioning of brassicapeptin in salt-stressed *A. thaliana* and Micro-Tom roots, irrespective of the 112 million years of reproductive isolation between these plants[43].

### Isolation and structural characterisation of R401 brassicapeptin

Next, we aimed to isolate the R401 brassicapeptin and elucidate its structure. Therefore, a 70 L fermentation of R401 was performed and extracted using ethyl acetate. The therefrom-resulting organic crude extract was further fractionated and purified using column chromatography (*e.g.*, medium pressure (flash) and high-performance liquid chromatography (HPLC) to finally yield the natural products brassicapeptin A as the major compound of this class, and brassicapeptin B in minor amounts. Additionally, the high-resolution electro spray ionization (HR-ESI)-MS/MS data analysis indicated two further minor derivatives, brassicapeptin C and D (Supplementary Note 1 and Supplementary Fig. 7). Brassicapeptin A and B were obtained as white amorphous powders and subsequently analysed by mass spectrometry (MS). The HR-ESI-MS spectrum of brassicapeptin A indicated a molecular weight of 2052.2004, suggesting a molecular formula of $C_{96}H_{161}N_{23}O_{26}$ (Supplementary Fig. 7a) and of $C_{94}H_{157}N_{23}O_{26}$ for brassicapeptin B ([M + 2H]$^{2+}$ *m/z* 1027.1080 and *m/z* 1005.0940, respectively) (Supplementary Fig. 7b). Comparison of the HR-ESI-MS/MS fragmentation patterns of brassicapeptin A and B gave first insights into the amino acid composition and revealed a high similarity between the molecules (*i.e.*, from $b_1$-$b_7$ and from $y_1$-$y_{10}$, except for differences from $b_8$-$b_{11}$ and from $y_{11}$-$y_{14}$) (Fig. 4a,b). To fully resolve their structures, nuclear magnetic resonance (NMR) experiments and Marfey's analysis were performed (for details about structure

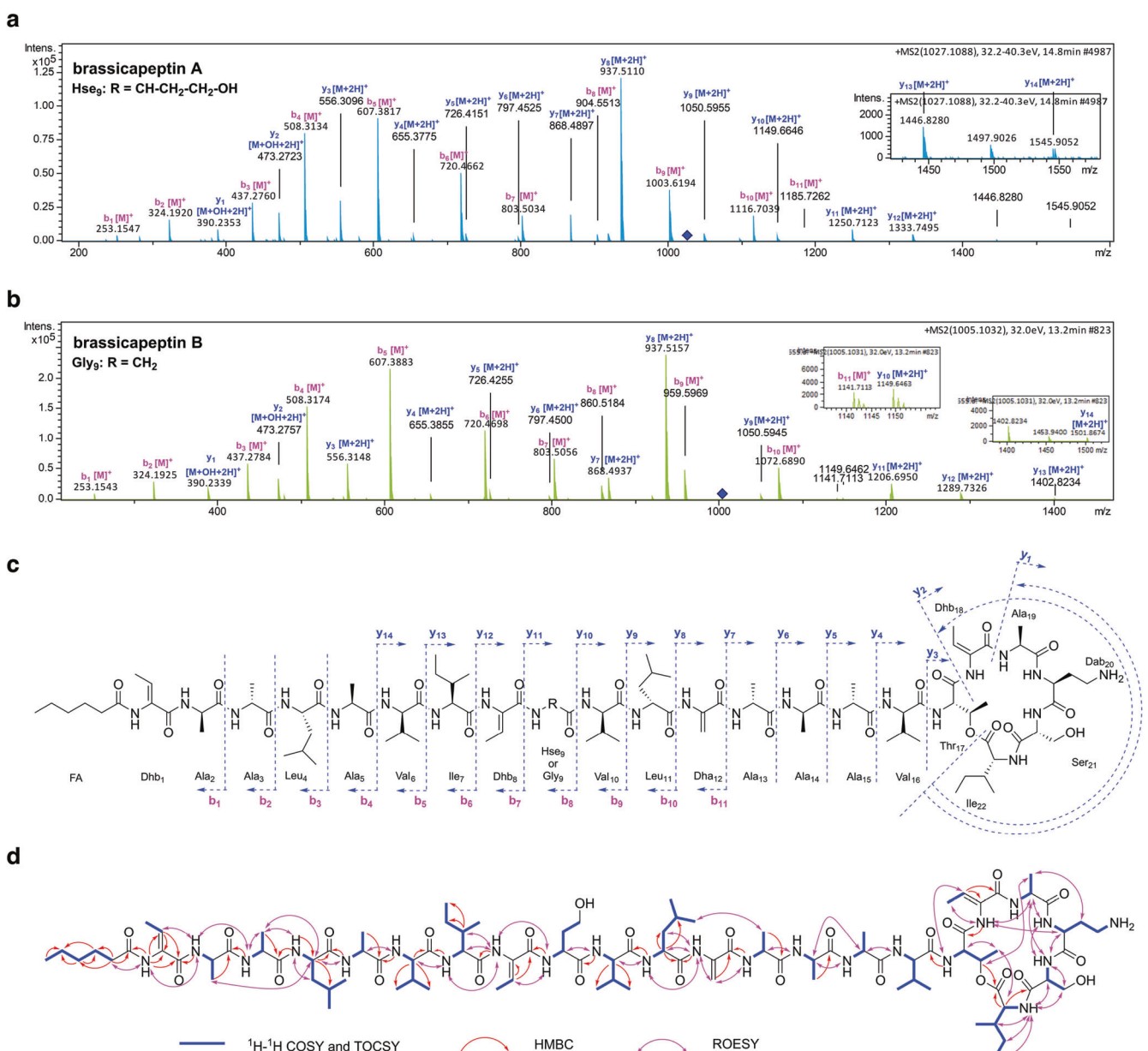

**Fig. 4 | Structure elucidation of R401 brassicapeptin A and B. a,b** HR-ESI-MS/MS fragments of brassicapeptin A (**a**) and brassicapeptin B (**b**). **c** Chemical structures of brassicapeptin A and B detected from ESI-MS/MS fragmentations and nuclear magnetic resonance (NMR) analysis. Brassicapeptin A and B differ only by one amino acid, which is a homoserine (Hse₉) in brassicapeptin A and a glycine (Gly₉) in B, respectively. **d** ¹H-¹H Correlated Spectroscopy (COSY; blue), Total Correlation Spectroscopy (TOCSY; blue), Heteronuclear Multiple Bond Correlation (HMBC; red) and Rotating-frame Nuclear Overhauser Effect Spectroscopy (ROESY; pink) revealed the complete structure of brassicapeptin A.

elucidation see Supplementary Note 1 and Supplementary Fig. 7). A combination of 1D and 2D experiments (¹H, ¹³C, HMBC, HSQC, COSY, TOCSY and ROESY) revealed the brassicapeptins to consist of 22 amino acid residues plus a fatty acid chain (Fig. 4c,d). The latter is six carbons in length, including a carbonyl group; the amino acid chain is cyclized by an intramolecular connection between threonine (Thr₁₇) and isoleucine (Ile₂₂) (Fig. 4c). Brassicapeptin A and B differ only by one amino acid, which is a homoserine (Hse₉) in brassicapeptin A and a glycine (Gly₉) in B, respectively (Fig. 4c). These newly discovered brassicapeptins represent large cyclic lipopeptides. Comparison to the previously described syringopeptins[44–46], which are also produced by *Pseudomonas* strains, revealed notable structural differences, including a different fatty acid starter unit and a smaller ring structure that is formed intramolecularly between the six C-terminal amino acid

residues (Supplementary Fig. 8); thereby suggesting that brassicapeptins represent a novel sub-group of cyclic syringopeptin-type lipopeptides. They also show high similarity to the reported cichopeptins and corpeptin[47,48], which possess a macrocycle formed by the five C-terminal amino acids. In brassicapeptins, this macrocycle is extended by one amino acid, ranging from the C-terminal Ile₂₂ towards Thr₁₇ (Supplementary Fig. 8).

**Brassicapeptin A and NaCl additively contribute to plant disease**

Because R401 is detrimental on *A. thaliana* grown on ½ Murashige-Skoog (½ MS) agar plates[6] (Fig. 1a), we first tested whether *brpC* also drives R401 pathogenicity in this system. Inoculation of R401 WT and *ΔbrpC* mutant, followed by shoot fresh weight measurements (14 dpi) revealed that *brpC* contributes, yet only partially, to R401 detrimental

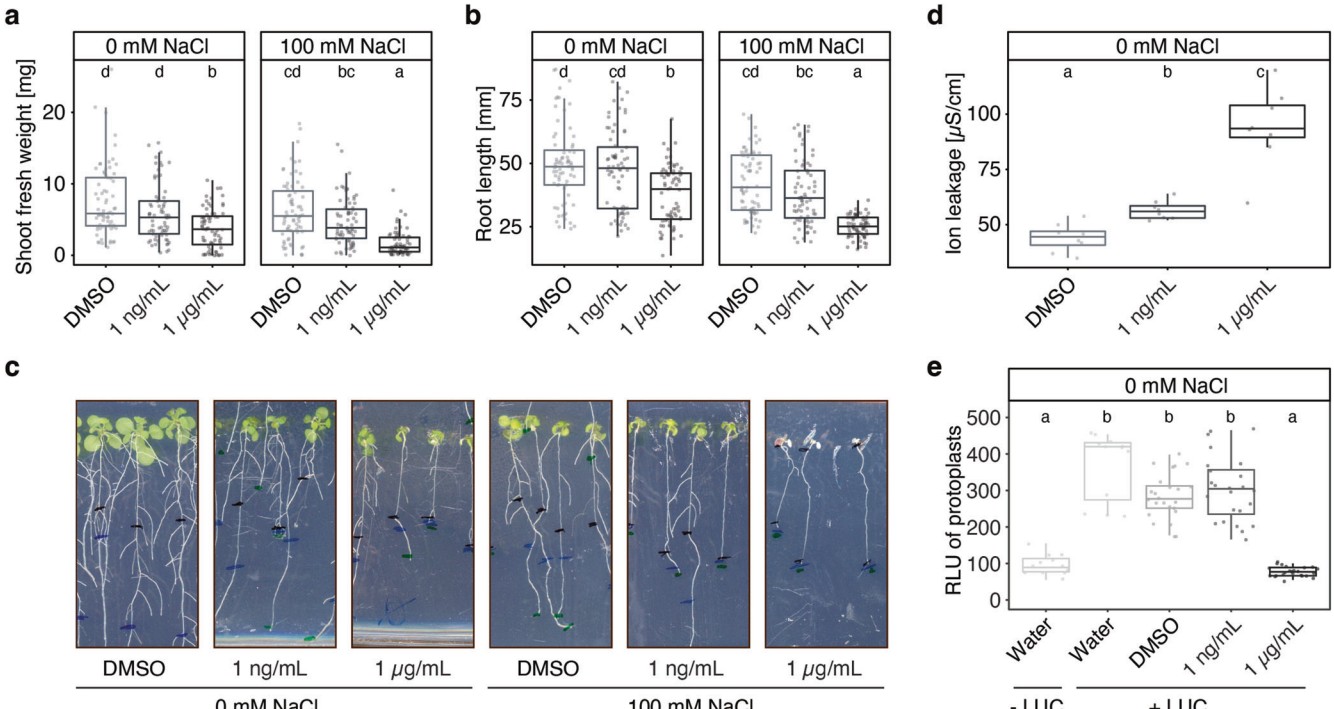

**Fig. 5 | Interaction between brassicapeptin A and NaCl promotes plant death.**
**a–c** Shoot fresh weight (**a**), root length (**b**) and images of representative phenotypes (**c**) of *A. thaliana* plants grown axenically on ½ MS agar plates supplemented with increasing concentration of brassicapeptin A and either 0 or 100 mM NaCl ($n = 72$ plants). Sterile *A. thaliana* seeds were pre-germinated on ½ MS agar plates for seven days and then transferred to new plates containing NaCl and/or brassicapeptin A for another 14 days; black, blue, and green markings in (**c**) indicate root length 0, 7 and 14 days after seedling transfer, respectively. **d** Ion leakage assay of *A. thaliana* leaf discs ($n = 8$ samples each comprising 5 leaf discs), 16 h after treatment with increasing concentrations of brassicapeptin A. DMSO or brassicapeptin A was taken up in sterile miliQ water. All solutions were measured before the experiment resulting in measurements of 2 μS/cm. **a–d** Brassicapeptin was solubilized in DMSO. Concentrations indicate final brassicapeptin A concentrations in the agar or miliQ water. **a, b, d** Letters indicate statistically significant differences as determined by Kruskal-Wallis followed by Dunn's post-hoc test and Benjamini-Hochberg adjustment with $p < 0.05$. Statistical comparisons were conducted for each salt treatment separately. **e** Luciferase activity was measured in *Arabidopsis thaliana* protoplasts inoculated with water, DMSO, or two concentrations of Brassicapeptin A (1 ng/mL and 1 μg/mL). Protoplasts were transfected with a LUC reporter assay (+LUC) or not transfected (-LUC). All protoplasts were incubated with water, DMSO or Brassicapeptin A for 16 hours before measuring Luciferase activity. DMSO was used as a control since Brassicapeptin A is solubilized in DMSO. Letters indicate statistically significant differences between conditions ($n = 24$ biologically independent samples), as determined by Kruskall-Wallis followed by Dunn's *post-hoc* test with $p < 0.05$ and Bonferroni adjustment. (**a, b, d, e**) Boxplots show 25th percentile, median, and 75th percentile.

activity in this agar-based gnotobiotic system ($p < 0.05$), suggesting that other bacterial genetic determinants are necessary to drive full pathogenicity (Supplementary Fig. 9). We next assessed the putative phytotoxic activity of brassicapeptin A in this reductionist system. We transplanted seven-day-old *A. thaliana* seedlings to ½ MS agar plates containing increasing concentrations of purified brassicapeptin A solubilized in dimethyl sulfoxide (DMSO), in the presence or absence of 100 mM NaCl. Within 14 days, brassicapeptin A showed a concentration-dependent effect on root and shoot growth, suggesting that the molecule alone in the absence of salt stress is already sufficient to induce a stunted growth phenotype (Fig. 5a-c). This is consistent with the fact that *brpC* contributes to R401 detrimental activity in this gnotobiotic system in the absence of salt stress (Supplementary Fig. 9). However, plants exposed to 1 μg/mL brassicapeptin A and 100 mM NaCl died immediately after transfer, whereas those exposed to 1 μg/mL brassicapeptin A or 100 mM NaCl alone remained alive and did not show leaf bleaching and severely inhibited root growth phenotypes (Fig. 5a-c). Given the phytotoxic effect of brassicapeptin A, we further assessed its potential mode of action. We observed that brassicapeptin A not only induced ion leakage in *A. thaliana* leaf discs after 16 h of incubation ($p < 0.05$, Fig. 5d), but also compromised viability of *A. thaliana* cells based on protoplast transfection assays with a luciferase construct[49] ($p < 0.05$, Fig. 5e). Our results suggest that this cyclic lipopeptide likely inserts into host plasma membranes to disrupt ion homeostasis, which is consistent with earlier work on syringopeptin, a

structurally-related compound that functions as a pore-forming molecule[50–52]. Therefore, brassicapeptin A-induced disruption of ion homeostasis, combined with increased osmotic pressure in the root environment, are likely jointly contributing to R401-induced disease symptoms in salt-stressed plants. Taken together, our data support the hypothesis that brassicapeptin production does not only promote R401 colonization at roots but also enhanced host susceptibility to osmotic stress, thereby leading to disease.

### Brassicapeptin A displays moderate antimicrobial activity against microbes from different kingdoms of life

Given the antimicrobial activity of known cyclic lipopeptides[53], we hypothesised that brassicapeptin contributes to the previously described remarkable inhibitory activity of R401[24]. We used the Δ*brpC* mutant and also generated a novel R401 triple mutant impaired in the production of pyoverdine, DAPG, and brassicapeptin (Δ*pvdY*Δ*phlD*Δ*brpC*, Fig. 6a) in order to abolish the dominant inhibitory activity of DAPG and pyoverdine, which co-explained >70% of R401 antagonistic activity based on previous measurements of R401 inhibitory halos[24]. Using eight microbes belonging to different taxonomic groups of the core root microbiota[4,54] (Fig. 6b), we observed— using the Δ*pvdY*Δ*phlD* mutant background—that *brpC* mildly contributed to microbial growth inhibition for half of the tested target bacterial and fungal strains (R83D2, R16D2, R31D1, F80, $p < 0.05$). The inhibitory activity of purified brassicapeptin A was further assessed

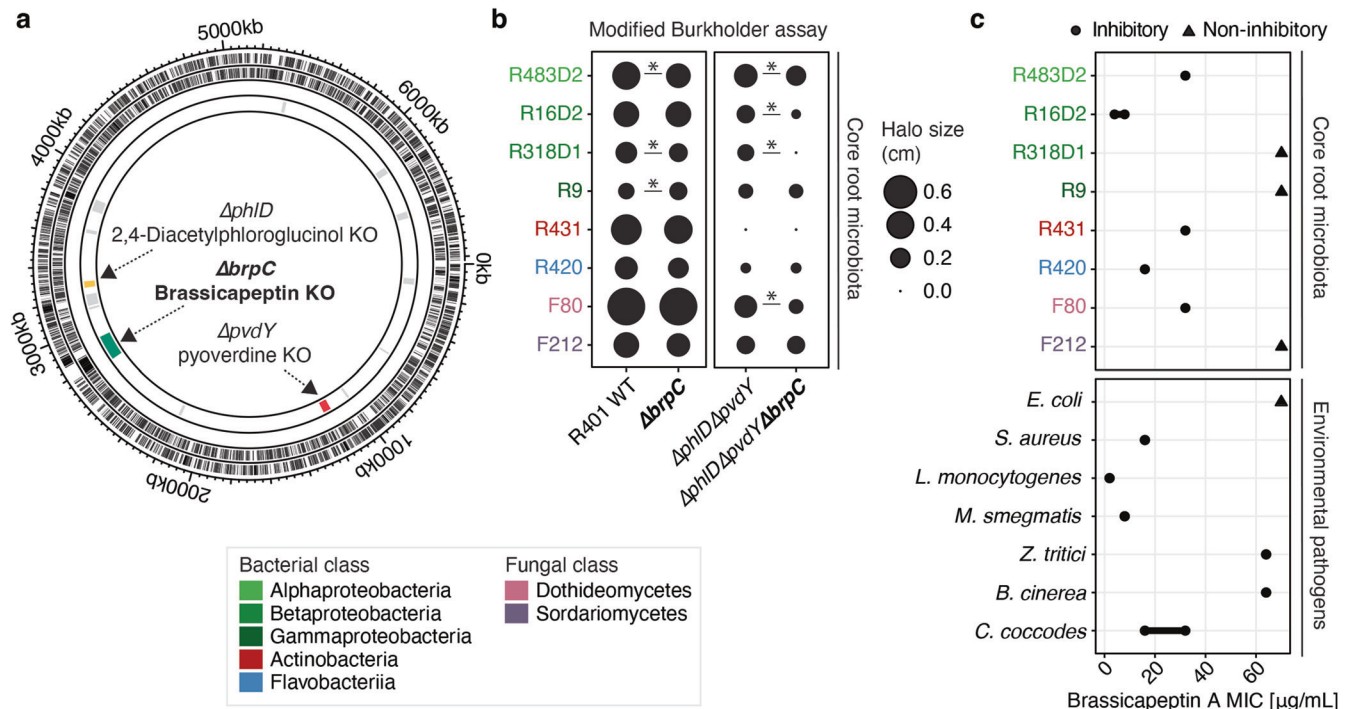

**Fig. 6 | Inhibitory activity of brassicapeptin A against phylogenetically diverse microbes. a** Genomic map of the main chromosome of R401, illustrating the locations of three biosynthetic gene clusters (BGC) involved in brassicapeptin A (green), 2,4-Diacetylphloroglucinol (DAPG, yellow) and pyoverdine (red) production in R401. Further indicated are respective mutants for each BGC *(ΔbrpC, ΔphlD, ΔpvdY*, respectively). R401 *ΔphlD* and *ΔpvdY* have been previously characterized[24]. **b** Balloon plot depicting the inhibitory activity of R401 WT, *ΔbrpC* (single), *ΔphlDΔpvdY* (double), or *ΔphlDΔpvdYΔbrpC* (triple) mutants, against six taxonomically diverse bacteria and two fungi belonging to the core microbiota of *A. thaliana*. Inhibitory activity was measured as halo of inhibition size (cm) by a modified Burkholder assay as described before[24]. '.' indicates no detectable halo

formation for the respective interaction. Statistical significance was determined by Wilcoxon test between R401 WT and *ΔbrpC* and between *ΔphlDΔpvdY* and *ΔphlDΔpvdYΔbrpC* ($p < 0.05$, $n = 9$ biologically independent samples). Target strains are coloured based on the bacterial or fungal classes. **c** Minimum inhibitory concentrations (MIC) of purified brassicapeptin A against the same core root microbiota of *A. thaliana* and environmental pathogens. Micro-broth-dilution assays were carried out to test a triplicated dilution series of brassicapeptin A (64−0.03 μg/mL). The potency of the compound is depicted as black circles (•) if an inhibitory effect could be observed, while black triangles (▲) indicate that no inhibition was observed (>64 μg/mL, see also. Supplementary Data 3).

against the same eight target strains as well as seven other environmental pathogens by micro-broth-dilution assays to determine the minimum inhibitory concentration (MIC) (Fig. 6c and Supplementary Data 3). Brassicapeptin A MICs displayed the lowest values for bacterial target strains *Listeria monocytogenes* DSM20600 (2 μg/mL), R16D2 (8-4 μg/mL), *Mycobacterium smegmatis* ATCC607 (8 μg/mL), R420 (16 μg/mL), *Staphylococcus aureus* ATCC25923 (16 μg/mL), R483D2 (32 μg/mL), R431 (32 μg/mL), and for fungal target strains *Colletotrichum coccodes* DSM62126 (32-16 μg/mL) and F80 (32 μg/mL), thereby validating the weak, yet multi-kingdom antimicrobial activity of this cyclic lipopeptide (Fig. 6c and Supplementary Data 3). Finally, we assessed the prevalence of the *syp-syr* containing BGC in plant-associated *Pseudomonas* strains by scrutinizing the genomes of several commensal culture collections encompassing 1,567 isolates retrieved from roots or leaves of *A. thaliana* or roots of the legume host *Lotus japonicus*. We observed that the *syp-syr* containing BGC is rare in these *Pseudomonas* genomes[2,5,55,56], contrasting with the higher prevalence of BGCs involved in pyoverdine and DAPG biosynthesis (Supplementary Fig. 10). Our results indicate that acquisition of this rare BGC by Pseudomonas isolates has nonetheless likely broad implication, not only for root colonization, disruption of host ion homeostasis and plant health under osmotic stress, but also for strain competitiveness via direct microbe-microbe competition.

## Discussion

Here, we report that salt-mediated dampening of host innate immunity likely contributes little to R401-mediated disease in soil-grown plants. Instead, we identify a bacterial exometabolite−with predicted pore-

forming activity−that is sufficient to drive disease in salt-treated plants and that dominantly functions, irrespective of the presence or absence of microbial competitors in soil. We report that brassicapeptin production drives root colonization and enhances susceptibility to osmotic stress, thereby contributing to disease emergence of soil-grown plants facing salt stress.

After 28 days of chronic salt treatment, the response of *A. thaliana* to salt stress was largely shoot-specific with photosynthesis-related genes being extensively downregulated, likely due to accumulation of Cl⁻ and Na⁺ ions in leaves that disrupt photosynthetic machinery[32,57,58]. In roots, only subtle transcriptional reprogramming was observed after 28 days of continuous salt stress, which can be explained by previous observation reporting a gradual decline of salt stress response over time in *A. thaliana* roots[34]. The effect of R401 in the absence of salt stress was also minor in both shoots and roots, while in the presence of hyperosmotic NaCl concentrations and of R401, extensive and root-specific transcriptional reprogramming was observed. This suggests that host response to disease induced by the combinatorial presence of salt and R401 primarily occurs in roots. Given that salt stress-induced dampening of immunity occurs in leaves, but not in roots, we propose that R401-induced stunted plant phenotypes under salt stress is largely immunity-independent. This is corroborated by our observation that R401 does not become detrimental on an immunocompromised *rbohD A. thaliana* mutant and by the fact that purified brassicapeptin A and salt are sufficient to induce disease-like symptoms in R401-free plant growth assays. Furthermore, low PAR treatment was shown to dampen SA- and JA-dependent immunity sectors in *A. thaliana* roots and shoots, which promotes

infection by *Botrytis cinerea* and *Pseudomonas syringae pv. tomato* DC3000[42]. However, R401 did not cause disease under such conditions. Although we cannot fully exclude the possibility that salt-induced dampening of immunity still contributes to R401 infection, our results suggest that this effect remains marginal.

R401 encodes a 140 kb BGC that is responsible for the production of the non-ribosomal peptides brassicapeptin A and B and the corresponding brassicamycin; however, only brassicapeptins could be detected in culture extracts. A R401 mutant lacking *brpC*, one of the core biosynthetic genes required for the production of brasscapeptin, showed impaired root colonization whilst retaining its WT-like growth in liquid media. In *A. thaliana*, impaired root colonization of the mutant was observed largely independently from the salt stress condition, indicating that *brpC* is a root colonization determinant. This same mutant also lost its ability to cause disease on salt-stressed plants in the FlowPot gnotobiotic system, demonstrating a requirement of brassicapeptin production for the detrimental activity of R401 in salt-treated plants. However, we report that disease symptoms caused by the combinatorial presence of R401 and salt stress was not associated with increased bacterial abundance at roots, suggesting that bacterial load is not causally linked to disease appearance in salt-stressed *A. thaliana*. Notably, structurally related compounds have been demonstrated to intercalate plasma membranes, thereby destabilising them and causing pore formation disrupting cell integrity, eventually leading to uncontrolled diffusion of cell solutes into the surrounding medium[45,59,60]. Although we could not test for similar effects of brassicapeptin in roots due to technical constraints, we nonetheless observed leakage of cellular solutes from *A. thaliana* leaf discs that were treated with brassicapeptin A, as well as brassicapeptin A-induced death of *A. thaliana* leaf protoplast, supporting the assumption that the molecule inserts into plant plasma membranes. Given that brassicapeptin production is needed for R401 proliferation at roots, we speculate that brassicapeptin promotes nutrient leaching from the root endosphere that fuel bacterial growth. Our results also suggest that brassicapeptin-induced ion homeostasis disruption is more damageable for plants facing high osmotic stress, thereby leading to disease. Consistently, we observed a linear dose-response relationship between applied NaCl concentrations and the detrimental effect of R401 on *A. thaliana*. We also showed *brpC*-dependent detrimental activity occurs under both salt and drought stress, but not under light stress, confirming that co-occurrence of hyperosmotic conditions and R401 brassicapeptin is required for disease in soil-grown plants. Finally, we report that brassicapeptin production is salt-inducible, at least in vitro. Altogether, these data indicate that plants colonized by R401 are more susceptible to salt stress than germ-free plants, which is also consistent with a root-specific transcriptional reprogramming observed after combinatorial treatment with R401 and NaCl.

Unlike in the FlowPot system, R401 was already detrimental for *A. thaliana* growth in the absence of salt stress in an ½ MS agar-based gnotobiotic plant system. In agar plates, the R401 mutant lacking *brpC* became less pathogenic than the WT strain but nonetheless retained some detrimental activity, indicating that *brpC* only partially contributes to R401 pathogenicity in this system. Our observation that *brpC* is contributing to R401 pathogenicity in agar plates is corroborated by the fact that purified brassicapeptin A was sufficient to negatively impact shoot and root phenotypes already in the absence of salt stress. Therefore, nutrient concentrations in ½ MS agar medium are likely sufficient to impose hyperosmotic stress on brassicapeptin A-treated plants. Consistent with our earlier observation showing that R401 *brpC* and salt are both needed to promote disease, we demonstrated that co-treatment with purified brassicapeptin A and 100 mM NaCl exacerbated the effect observed for brassicapeptin A alone, thereby leading to dead plants. Collectively, this indicates that the interaction between a specialized bacterial exometabolite and high osmotic concentrations in the environment is sufficient to explain R401 detrimental activity on salt-sensitive plants.

R401, R131 and R107 have been isolated from healthy *A. thaliana* plants; however, under favourable conditions they can become detrimental to plant health – in the case of R401 even in the context of natural or synthetic microbial communities. Groundwater-derived *Pseudomonas sp.* N2C3 (N2C3) also contains the *syp-syr* BGC and has been demonstrated to cause *syp*-dependent stunting of *A. thaliana* root and shoot growth in ½ MS agar medium[61]. In natural soil, N2C3 does not cause any detrimental phenotypes, even after inoculation of high bacterial titres[62] ($1 \times 10^6$ cells per gram soil, a phenotype that is reminiscent of the herein described effects of R401). Using computational analyses, convergent gain and loss of the *syp-syr* BGC has been demonstrated for the *Pseudomonas fluorescence*-clade, which comprises *P. brassicacearum*[61]. Loss of *brpC* in R401 was sufficient to turn this detrimental strain into a plant growth-promoting strain under salt stress. While the mechanisms of growth promotion of the *ΔbrpC* R401 mutant remain elusive, it is conceivable that the acquisition of the *syp-syr* BGC in this strain – and likely in other *Pseudomonas* spp. isolates - might overwrite their plant growth-promoting capabilities.

High soil salinity is one of the main constraints for agricultural performance worldwide and arises through frequent irrigation and fertilisation, which results in the accumulation of nutrient salts in agricultural soils. This accumulation will become more problematic due to an increasing demand for field irrigation owing to climate change[33,35,63–65]. Our data provide first line of evidence indicating that the interplay between the host, its microbiota, and the osmotic environment can conditionally lead to disease due to the presence of specific bacterial taxa that employed exometabolites that likely disrupt ion homeostasis and promote nutrient leakage at the host interface. Given that the disease phenotype conferred by R401 under salt stress is retained in a microbial community context and in the natural CAS soil, it reflects an important phenomenon that has physiological relevance for plant disease emergence in natural environment. While R401 likely fulfils extensive biocontrol activities due to its diverse repertoire of antibacterial and antifungal specialized exometabolites[24], favourable abiotic conditions allow for disease development by brassicapeptin-producing R401. Our results indicate that acquisition of brassicapeptin production capability has likely provided a competitive advantage for R401 not only for root colonization but also for microbe-microbe competition. Therefore, we delineate how a single bacterial molecule can have multiple independent effects on organisms that evolved in at least three kingdoms of life (plants, bacteria, fungi), thereby contributing to bacterial competitiveness at roots and promoting plant susceptibility to osmotic stress. It is also tempting to speculate that membrane-intercalating exometabolites outside of the genus *Pseudomonas*, such as surfactin produced by *Bacillus spp.* may cause similar detrimental activity as R401 brassicapeptin A[66–70]. Taken together, our data provides a mechanistic explanation for the emergence of a disease in the plant microbiome that requires a single bacterial exometabolite and adequate abiotic stress conditions. Our work also defines an ecological framework to understand the conditional detrimental activity of R401 and likely other *Pseudomonas spp.* isolates in complex soil environments.

## Methods

### Primers
All primers used in this study can be found in Supplementary Data 4.

### Microorganisms
The bacterial and fungal strains used in this study have been initially isolated from unplanted soil, *A. thaliana* roots or shoots[2,3] and are summarized in Supplementary Data 5. The R401 *ΔbrpC* mutant has been deposited in the bacterial culture collection of the Department of

Plant Microbe Interactions at the Max Planck Institute for Plant Breeding Research in Cologne, Germany, and are available upon request from Stéphane Hacquard (hacquard@mpipz.mpg.de).

## Plant species

*A. thaliana* ecotype Columbia-0 (Col-0), *Lotus japonicus* ecotype Gifu B-129, and *Solanum lycopersicum cv*. Micro-Tom were used as wildtypes. *A. thaliana rbohd* contains a *dSpm* transposon in the fifth exon of *AtrbohD*[71] (AT5G47910)

## Microbial culture conditions

Bacteria were streaked from glycerol stocks (25% glycerol) on TSA plates (15 g/L Tryptic Soy Broth, Sigma Aldrich; with 10 g/l Bacto Agar, Duchefa Biochemie) and grown at 25 °C. Single colonies were inoculated in liquid 50% TSB (15 g/L Tryptic Soy broth, Sigma Aldrich) and grown until dense at 25 °C with 180 rpm agitation. Dense cultures were then stored at 4 °C and diluted 1 to 10 in TSB the day before the experiment and cultured at 25 °C with 180 rpm agitation overnight to ensure sufficient cell densities for slow- and rapidly-growing bacteria. Glycerol stocks were stored at -80 °C and kept on dry ice when transported. Individual pieces of fungal mycelium were transferred to potato dextrose agar (PDA; Sigma-Aldrich) Petri dishes from glycerol stocks (approx. 30 pieces of fungal mycelium in 25% sterile glycerol, stored at -80 °C). Fungi were grown at 25 °C in the dark for 14 days.

## Seed sterilisation

*A. thaliana* and *S. lycopersicum* Micro-Tom seeds were sterilized using 70% ethanol and bleach. Seeds were submerged in 70% ethanol and left shaking at 40 rpm for 14 minutes. Ethanol was removed before the seeds were submerged in 8.3% sodium hypochlorite (Roth) containing 1 μL of Tween 20 (Sigma-Aldrich) and left shaking at 40 rpm for 4 minutes. Under sterile conditions, the seeds were washed 7 times and finally taken up with sterile 10 mM $MgCl_2$. Seeds were left for stratification at 4 °C for 3 days. Seed sterility was confirmed by plating approx. 100 seeds on a 50% TSA plate. The seed coat of *L. japonicus* Gifu seeds was first abraded using sanding paper, then seeds were incubated for 20 min in diluted bleach, followed by five-times washing in sterile water. Sterilized Gifu seeds were pregerminated on sterile, water-soaked Whatman paper for 7 days.

## Gnotobiotic Flowpot experiments

Flowpot assembly was performed according to Kremer and colleagues[26] with minor adjustments[24]. A 2:1 mixture of peat potting mix and vermiculite was used as a matrix. The matrix was sterilized two times (25 min liquid cycle (121 °C) and 45 min solid cycle (134 °C)) and stored at 60 °C until completely dry. Prior to Flowpot assembly, the matrix was rehydrated with sterile MiliQ water. Flowpots were assembled by adding a layer of glass beads to the conical end of a truncated syringe, followed by a layer of the rehydrated, sterile substrate, subsequently covered with a sterile mesh secured by a cable tie. Assembled Flowpots were sterilized on a 25 min liquid cycle, stored at 60 °C overnight and sterilized twice on a 45 min solid cycle. For Micro-Tom and Gifu, big Flowpots fitting 50 mL soil were used, as described by Wippel and colleagues[55]. Microbes were grown and inocula were prepared as described above. Each Flowpot was inoculated with 50 mL half-strength Murashige and Skoog medium with vitamins (½ MS; 2.2 g/L, Duchefa Biochemie, 0.5 g/L MES, pH 5.7). For bacteria, a final $OD_{600}$ of 0.0025 in 50 ml ½ MS were inoculated per Flowpot. For salt stress treatment ½ MS contained 50, 100 or 150 mM NaCl. For drought treatment, ½ MS contained 5% polyethylene glycol (PEG8000; Sigma-Aldrich). Per Flowpot, five or three surface-sterilized and stratified *A. thaliana* or Micro-Tom seeds were inoculated, respectively. For Gifu, 7 days old, pregerminated seedlings with similar developmental stages were carefully transferred to the Flowpots. Microboxes were then incubated in a light cabinet under short day conditions (10 h light at

21 °C, 14 h dark at 19 °C) for 28 days and randomized every 2–3 days. For low PAR treatments, microboxes were partly covered in cardboard boxes, as described in Hou et al.[42], with a photosynthetic photon flux density of 27.91 μmol m$^{-2}$ s$^{-1}$. The light condition was measured by Spectral PAR meter PG100N (UPRtek).

## Natural soil experiments

Cologne agricultural soil (CAS) was obtained from the Max Planck Institute for Plant Breeding Research in Cologne, Germany. R401 WT was cultured as described above. CAS was placed in squared pots with an edge length of 9 cm and flush inoculated with 100 mL sterile water or 100 mM NaCl containing either live or heat-killed R401 WT cells at an $OD_{600}$ of 0.0025. Four surface-sterilized and stratified Col-0 seeds were placed per pot. Pots where then placed in trays in the greenhouse for 28 days. The temperature was set at 22 °C during day and 18 °C during night, with a relative humidity at 65% and 16 h of light.

## Agar plate experiments

For the experiment depicted in Fig. 1a and Supplementary Fig. 9, 24 surface-sterilized and stratified *A. thaliana* seeds were placed in two rows per 12 cm square plate containing ½ MS medium with 10 g/L Bacto-Agar (Duchefa Biochemie). Agar plates were sealed and incubated in a light cabinet under short day conditions (10 h light at 21 °C, 14 h dark at 19 °C) for 14 days. At day 14, plants were flushed with 15 mL 10 mM $MgCl_2$ containing either live or heat-killed R401 WT cells at an $OD_{600}$ of 0.0005 for 5 mins. Plants were transferred to new plates and grown for another 5 days for a total of 19 days. Shoot fresh weight was measured as a proxy to determine bacterial detrimental activity.

## RNA Seq experiments

Total RNA was extracted from *A. thaliana* roots and shoots by RNeasy Plant Mini Kit (Qiagen). RNA-Seq libraries were prepared by the Max Planck Genome-centre Cologne with NEBNext® Ultra™ II Directional RNA Library Prep Kit for Illumina® and then sequenced on a NextSeq 2000 in 2 ×150 paired-end read mode. RNA-Seq read quality was observed with FastQC v0.11.9, then reads were trimmed with Trimmomatic PE v0.38[72] using parameters TRAILING:20 AVGQUAL:20 MINLEN:100. Trimmed reads were then mapped on the reference *A. thaliana* genome TAIR10 using Hisat2 v2.2.1[73], taking into consideration exon and splicing sites locations (according to annotation file TAIR10_GFF3_genes.gff downloaded from arabidopsis.org in October 2022). The number of fragments (pair of reads) mapped on each gene was then quantified using featureCounts v2.0.0[74] (parameter -p, default settings). Resulting data were used to calculate FPKM (fragments per kilobase of transcript per million fragments mapped) values for each gene in each sample: (1) Scaling factor: SF=Total number of mapped reads / 1e6; (2) Fragments per million: FPM=Number of reads mapped on one gene / SF; (3) FPKM = FPM / (Gene length / 1000). Numbers of mapped reads on each gene were also used to perform differential gene expression analysis with DESeq2 v1.24.0[75] and functions estimateSizeFactor, estimateDispersions and nbinomWaldTest. log2FoldChanges values were then corrected with shrinkage algorithm apeglm v1.6.0[76]. One R401 WT and 100 mM NaCl-treated shoot sample (sample ID: 5642.W) was highly contaminated with brown trout (*Salmo trutta*) reads, likely arising during library preparation. This sample was therefore excluded from the analysis.

## antiSMASH

antiSMASH[40] predictions are derived from Getzke et al.[24]. Bacterial genomes were downloaded from "www.at-sphere.com" or NCBI and submitted to https://antismash.secondarymetabolites.org/ version 6.0. Only high-quality genomes, as assessed by CheckM with ≥90% completeness and ≤5% contamination ratio were used for the analysis. For R401, the PacBio-sequenced high-quality genome was used for BGC prediction using antiSMASH.

## ΔbrpC mutant generation

R401 ΔbrpC mutant generation was conducted as described in Getzke.[24] and Vannier et al.[77]. All utilized primers can be found in Supplementary Data 4. Marker-free knockouts in R401 were generated through homologous recombination using the cloning vector pK18mobsacB (GenBank accession: FJ437239), which encodes the kanR and sacB genes conferring resistance to kanamycin and susceptibility to sucrose, respectively. In this method, upstream and downstream sequences of the gene to be deleted are integrated into the pKl8mobsacB suicide plasmid by Gibson assembly[78]. The resulting plasmid is transformed into BW29427 E. coli cells and subsequently conjugated into R401. The plasmid is then integrated into the chromosome by homologous recombination and deletion mutants are generated by a second sucrose counter-selection-mediated homologous recombination event[79].

**Generation of pK18mobsaB-derived plasmid containing flanking regions of the gene of interest.** Primers were designed to amplify a 750-bp DNA sequence (i.e., flanking region) directly upstream and downstream of the target region, sharing terminal sequence overlaps to the linearized pK18mobsacB vector and the other respective flanking region using Geneious Prime. R401 genomic DNA was isolated from 6 μl dense R401 culture in 10 μl of buffer I (pH 12) containing 25 mM NaOH, 0.2 mM EDTA at 95 °C for 30 min, before the pH was readjusted using 10 μl of buffer II (pH 7.5) containing 40 mM Tris-HCl. The R401 genomic DNA was used for amplification of the flanking regions through PCR using the respective flanking region-specific primer combinations (Supplementary Data 4). PCR was conducted with 0.2 μl Phusion Hot Start High-Fidelity DNA polymerase (New England Biolabs) in 20-μl reactions containing 4 μl 5x Phusion HF buffer (New England Biolabs), 0.4 μl 10 mM dNTPs, 1 μl of 10 μM forward primer, 1 μl of 10 μM reverse primer, 2 μl of R401 genomic DNA as template, filled up to 20 μl with nuclease-free water. The tubes were placed into a preheated (98 °C) thermal cycler set at the following program: 98 °C for 30 s, 35 cycles of 98 °C for 7 s, 60 °C for 20 s, 72 °C for 15 s, then a final extension at 72 °C for 7 min. Five microliters of the PCR product were combined with 1 μl Orange DNA Loading Dye (6x; New England Biolabs), loaded on 1% agarose gels containing 0.05% EtBr, and run at 110 mV. After confirmation of successful amplification, the PCR product was purified using AMPure XP (Beckman-Coulter) and subsequently quantified using Nanodrop (Thermo Fisher Scientific). Plasmid purification was performed on an E. coli culture containing plasmid pK18mobsacB using the QIAprep Spin Miniprep Kit for plasmid DNA purification (QIAGEN) following the manufacturer's instructions. The pkl8mobsacB vector was then amplified and linearized through PCR using the pk18mobsac_F (PKSF) and pk18mobsac_R (PKSR) primers (Supplementary Data 4). PCR was conducted with 0.2 μl Phusion Hot Start High-Fidelity DNA polymerase (New England Biolabs) in 20-μl reactions, largely as described above with 1 μl 0.1 ng/μl pkl8mobsac as a template. Annealing temperature was decreased to 55 °C and extension time increased to 150 s for each cycle. Template DNA was digested by DpnI (New England Biolabs) in 50-μl reactions containing 1 μl DpnI, 1 μg DNA, 5 μl Cutsmart buffer (New England Biolabs) and filled up to 50 μl with nuclease-free water. The tubes were then incubated at 37 °C for 15 min followed by heat inactivation at 80 °C for 20 min. Five microliters of the DpnI-digested plasmid were combined with 1 μl Orange DNA Loading Dye and analysed by DNA agarose electrophoresis. Upon successful verification of amplification and digestion, the remaining sample was purified using AMPure XP and subsequently quantified using Nanodrop. Linearized pK18mobsacB and both flanking regions were mixed in a molar ratio of 1:3:3 into a 10-μl total volume, added to 10 μl 2X Gibson Assembly® Master Mix (New England Biolabs) and incubated at 50 °C for 1 h.

**Transformation into chemically competent E. coli BW29427 cells.** The vector was transformed into 50 μl chemically competent BW29427 E. coli cells according to the following heat shock protocol: 2 μl of the vector were gently mixed with 50 μl of competent cells, and the resulting mixture was incubated on ice for 30 min. The mixture was transferred to a water bath at 42 °C for 1 min and put back on ice for 2 min. Then, 1 ml of 50% TSB with 50 μg/ml diaminopimelic acid (DAP; Sigma-Aldrich) was added to the heat-shocked cells, the mixture was left to regenerate at 37 °C for 1 h and then plated on 50% TSA containing 25 μg/ml Kanamycin (Kan) and 50 μg/ml DAP. The plates were incubated at 37 °C overnight. The resulting colonies were validated by colony PCR using the M13F and M13R primers. Colony PCR was performed on at least four separate colonies with 0.4 μl DFS-Taq polymerase (BIORON) in 25 μl reactions containing 2.5 μl l0x incomplete buffer (BIORON), 0.5 10 mM MgC12, 0.5 μl 10 mM dNTPs, 0.75 μl 10 μM forward primer, 0.75 μl 10 μM reverse primer, a small fraction of a colony and filled up to 25 μl with nuclease-free water. The tubes were placed in a thermocycler set at the following program: 94 °C for 2 min, 35 cycles of 94 °C for 30 s, 55 °C for 30 s, 72 °C for 2 min, then a final extension at 72 °C for 10 min. Five microliters of the PCR product were combined with 1 μl Orange DNA Loading Dye and analysed by DNA agarose electrophoresis. Positive colonies were purified by streaking on new 50% TSA plates containing 25 μg/ml Kan and 50 μg/ml DAP and further verified by Sanger sequencing (Eurofins Scientific) following the manufacturer's protocol.

**Conjugation of E. coli and R401 and selection for first homologous recombination event.** E. coli BW29427 cells containing the plasmid and R401 were inoculated into 4 ml of 50% TSB containing 25 μg/ml Kan and 50 μg/ml DAP or 50% TSB and incubated overnight at 37 °C with 180 rpm agitation or 25 °C with 180 rpm agitation, respectively. Cells were harvested by centrifugation at 8000 rpm for 2 min at room temperature, then washed 3x and subsequently resuspended in 1 ml of 50% TSB followed by centrifugation, after which the supernatant was discarded. After quantifying OD$_{600}$, both cultures were mixed to equal parts and approx. 10x concentrated by centrifugation. The bacterial suspension was plated on 50% TSA plates containing 50 μg/ml DAP and incubated at 25 °C overnight to allow for conjugation events. The mating patches were scraped of the plate and resuspended in 1 ml 50% TSB. Then, 100 μl were spread on 50% TSA plates containing 25 μg/ml Kan and 50 μg/ml Nitrofurantoin (Nitro; Sigma-Aldrich; to counter-select E. coli) and incubated at 25 °C. Colonies were validated for successful genomic insertion of the plasmid via colony PCR using a primer specific to the genomic DNA approx. 150 bp upstream of the upward flanking region (upup) and the plasmid specific M13F primer. Colony PCR was performed on at least 15 separate colonies and a WT control with 0.4 μl DFS-Taq polymerase in 25-μl reactions as described previously, but with an annealing temperature of 60 °C. Five microliters of the PCR product were combined with 1 ml Orange DNA Loading Dye and analysed by DNA agarose electrophoresis followed by Sanger sequencing following the manufacturer's protocol.

**Sucrose counter-selection to induce the second homologous recombination event.** A R401 colony with a successful genomic insertion of the plasmid was resuspended from a plate into 1 ml of 50% TSB. The cell density in the medium was then measured using the Multisizer 4e Coulter Counter (Beckman Coulter) following the manufacturer's protocol. One hundred microliters of 500 cells/μl, 5,000 cells/μl and 50,000 cells/μl dilutions were spread on three separate 50% TSA plates containing 300 mM sucrose. The plates were incubated at 25 °C for approx. 48 h. At least 30 colonies were examined by colony PCR using the respective upup and dwdw primers. Colony PCR was performed with 0.4 μl DFS-Taq polymerase in 25-μl reactions as described previously with an annealing temperature of 60 °C. Five

microliters of the PCR product were combined with 1 µl Orange DNA Loading Dye and analysed by DNA agarose electrophoresis. Positive colonies were purified by streaking on new 50% TSA plates and further verified by Sanger sequencing (Eurofins Scientific) following the manufacturer's protocol. They were also streaked on 50% TSA containing 25 µg/ml Kan to verify loss of the plasmid. A second colony PCR was performed on positive colonies and a wt control to validate the absence of the GOI, using a forward (inF) and reverse (inR) primer inside the GOI. Colony PCR was performed with 0.4 µl DFS-Taq polymerase in 25 µl reactions as described previously. Five microliters of the PCR product were combined with 1 ml Orange DNA Loading Dye and analysed by DNA agarose electrophoresis. Upon successful verification, 4 ml of 50% TSB were inoculated with a positive colony and grown overnight at 25 °C at 180 rpm. Finally, 750 µl of the overnight culture were added to 750 µl of 50% glycerol in an internally threaded 1.8 ml Nunc CryoTube, gently mixed, and stored at -80 °C.

## Quantification of brassicapeptin production
Abolishment of brassicapeptin production in the deletion mutant was corroborated by HR-UPLC-MS measurement. Therefore, R401 WT and R401 $\Delta brpC$ were grown in 50% TSB for 3 days; subsequently samples were taken and analysed as described in Getzke et al.[24]. (Fig. 3b).

## Quantification of R401 load on plant roots and shoots
Col-0, Gifu and Micro-Tom roots and Gifu and Micro-Tom shoot (28 days old, grown in FlowPot gnotobiotic system, See Fig. 3e, Supplementary Fig. 1c and 6c,d) were carefully cleaned, dried and collected in pre-weighed, sterile 2 mL tubes containing 1 steel bead (3 mm diameter). Tubes were weighed again to assess the root or shoot fresh weight. Subsequently, samples were ground in a Precellys 24 Tissue-Lyser (Bertin Technologies) for 2 ×30 s at 6,200 rpm with 15 s intervals. Then, 150 µL of sterile 10 mM $MgCl_2$ were added to each tube and roots were ground again under the same conditions. The homogenate was serially diluted in 10 mM $MgCl_2$. Undiluted samples and each dilution were plated on 50% TSA square plates, dried and left to grow at 25 °C until single colonies appeared. Colonization was expressed as Log2 CFU per mg of roots or shoot. Pictures were taken and single colonies were counted blinded.

## Microbial growth rates validation
Assessment of microbial growth rates was conducted as described before[24]. Either artificial root exudates (ARE) or ARE supplemented with 100 mM NaCl were inoculated with R401 WT or $\Delta brpC$ cells to a final $OD_{600}$ 0.01.

## Isolation of R401 brassicapeptin
R401 was precultured in 300 mL flasks containing 100 mL TSB medium for 2 days at 30 °C and 160 rpm. 80 mL preculture were added to 1 L M19 medium (casein peptone 20 g/l, D-mannitol 20 g/L) in 5 L flasks. This procedure was carried out 70-times. All flasks were incubated at 30 °C and 160 rpm for 24 hours, followed by an extraction using EtOAc (volume ratio 1:1) for three times, yielding 16.24 g crude extract. Twenty-one fractions were collected from reversed phase flash chromatography (Interchim Puriflash 4125 chromatography system with Puriflash C18-AQ30 µm F0120 column) with an elution gradient starting from 10% MeOH/H$_2$O to 100% MeOH over 4 h. Fraction 19 (143.8 mg) was further subjected to semi-preparative HPLC (semi-preparative purification column: VP 250/10 Nucleodur C18 Gravity-SB, 5 µm; Macherey-Nagel, Flow: 3 mL/min; Gradient: 0–20 min, gradient increased from 40% to 100% MeOH; 20–32 min, 100% MeOH) to give two subfractions (fractions 19.1 and 19.2). Fraction 20 (96.7 mg) was also subjected to semi-preparative HPLC (semipreparative purification column: VP 250/10 Nucleodur C18 Gravity-SB, 5 µm; Macherey-Nagel; Flow: 3 mL/min; Gradient: 0–20 min, gradient increased from 40% to 100% MeOH; 20–32 min, 100% MeOH) to give two subfractions

(fractions 20.1 and 20.2). Subfraction 19.2 (6.7 mg) and 20.2 (6.7 mg) were further purified by semi-preparative HPLC (analysis column: EC 250/4.6 Nucleodur C18 Gravity-SB, 5 µm; Macherey-Nagel; Flow: 1 mL/min; Gradient: 0–40 min, gradient increased from 40% to 100% MeOH; 40–50 min, 100% MeOH) to yield brassicapeptin A (6.1 mg, $t_R$ = 38.4 min). Fraction 18 (108.6 mg) was subjected to semi-preparative HPLC (semipreparative purification column: VP 250/10 Nucleodur C18 Gravity-SB, 5 µm; Macherey-Nagel; Flow: 3 mL/min; Gradient: 0–20 min, gradient increased from 40% to 100% MeOH; 20–32 min, 100% MeOH) to give three subfractions (fractions 18.1–18.3). Subfraction 18.3 (8 mg) was further purified by semi-preparative HPLC (analysis column: EC 250/4.6 Nucleodur C18 Gravity-SB, 5 µm; Macherey-Nagel; Flow: 1 mL/min; Gradient: 0–3 min, 10% MeCN; 3–58 min, gradient increased from 10% to 92.5% MeCN; 58–65 min, 100% MeCN) to yield brassicapeptin A (3 mg, $t_R$ = 56.4 min) and B (0.9 mg, $t_R$ = 52.3 min).

## Structure elucidation of R401 brassicapeptin
The planar structure of the isolated compounds was elucidated by analysis of NMR data, LC-HR-MS and LC-HR-MS/MS data. The 1D and 2D NMR spectra were recorded in $CD_3OD$ or DMSO-$d_6$ using Bruker Avance II 600 MHz spectrometers equipped with a Prodigy cryoprobe (Bruker, Ettlingen, Germany) and Bruker Avance Neo 700 MHz spectrometer equipped with a 5 mm CryoProbe Prodigy TCI ($^1$H,$^{15}$N,$^{13}$C Z-GRD) (Bruker). The NMR data can be found in Supplementary Data 6, all 1D and 2D NMR spectra can be found in Supplementary Fig. 7c-i. The LC-HR-MS and MS/MS data were recorded on a quadrupole time-of-flight spectrometer (LC-QTOF maXis II, Bruker Daltonik) equipped with an electrospray ionization source in line with an Agilent 1290 infinity LC system (Agilent). C18 RP-UHPLC (ACQUITY UPLC BEH C18 column; 130 Å, 1.7 µm, 2.1 × 100 mm) was performed at 45 °C with the following linear gradient: 0 min: 95% A; 0.30 min: 95% A; 18.00 min: 4.75% A; 18.10 min: 0% A; 22.50 min: 0% A; 22.60 min: 95% A; 25.00 min: 95% A (A: H$_2$O, 0.1% HCOOH; B: CH3CN, 0.1% HCOOH; flow rate: 0.6 mL/min). Mass spectral data were acquired using a 50 to 2,000 $m/z$ scan range at 1 Hz scan rate. MS/MS experiments were performed with 6 Hz and the top five most intense ions in each full MS spectrum were targeted for fragmentation by higher-energy collisional dissociation at 25 eV or 55 eV using N2 at 10−2 mbar. Precursors were excluded after two spectra, released after 0.5 min, and reconsidered if the intensity of an excluded precursor increased by factor 1.5 or more. The HR-ESI-MS data can be found in Supplementary Fig. 7a,b,k,l, the HR-ESI-MS/MS data can be found in Supplementary Data 7. The absolute configuration of isolated compounds was elucidated by Marfey assay. A 5 mM stock solution in H$_2$O was prepared from the reference amino acids. 20 µL 1 M NaHCO$_3$ and 50 µL 7 mM L FDVA (Sigma Aldrich) in acetone was added to 50 µL stock solution of the reference amino acids. The mixture was stirred at 40 °C for 3 h and then quenched by adding 20 µL of 1 M HCl. After evaporation, the residue was dissolved in 40 µl DMSO and analysed by UPLC HRMS (maXis II). brassicapeptin A (0.5 mg) and B (0.3 mg) were dissolved in 200 µL of 6 M DCl in D$_2$O and stirred at 160 °C for 7 h. After concentrating the solution under reduced pressure, the residue was dissolved in 200 µl H$_2$O, and 100 µL of 1 M NaHCO$_3$ and 200 µL of 7 mM L FDVA in acetone were added. After stirring for 3 h at 40 °C, the solution was quenched by adding 100 µL of 1 M HCl. After evaporation to dryness, the residue was dissolved in 50 µL DMSO and analysed by UPLC HRMS (maXis II). The results of the Marfey analysis can be found in Supplementary Fig. 7j.

## In planta activity test of R401 brassicapeptin A
Surface sterilized and stratified *A. thaliana* seeds were pregerminated on ½ MS agar plates. After seven days, seedlings were transferred to ½ MS agar plates supplemented with either 1 ng/µL, 1 µg/µL

brassicapeptin A solubilized in DMSO or DMSO as negative control and either 0 mM or 100 mM NaCl. After 14 days agar plates for seven days and then transferred to new plates containing NaCl and/or brassicapeptin A. Agar plates were incubated in a light cabinet under short day conditions (10 h light at 21 °C, 14 h dark at 19 °C) for additional 14 days and randomized every 2–3 days.

### Ion leakage assay
Five discs with 3 mm diameter of approx. 28 days old *A. thaliana* Col-0 leaves were transferred to wells of a 24-well plate, filled with sterile MiliQ water supplemented with either 1 ng/μl, 1 μg/μL brassicapeptin A solubilized in DMSO or DMSO as negative control. Before the transfer of leaf discs and after 16 h, ion leakage measurements were taken using the Twin Cond conductivity meter B-173 (HORIBA).

### Protoplast transfection assay
*Arabidopsis thaliana* Col-0, were grown on ½ MS agar for 2 weeks at 22 °C during the day (16 hours) and 18 °C during the night (8 hours). Protoplasts were isolated from leaves and transfected following a protocol adapted from[49]. Briefly, leaves were chopped and mixed with an enzyme solution (1.5% Celulase R10 and 0.5% Maceroenzymes R10) for 3 hours in darkness, then filtered through a 100 μm nylon mesh. The concentration of protoplasts was estimated using a hemocytometer and adjusted to $5.10^5$ cells/mL. Protoplast solutions (300 μL of $5.10^5$ cells/mL) were transfected with 6 μg of the luciferase (LUC) reporter construct (pZmUBQ:LUC)[49]. Transfected protoplasts were treated with either water or DMSO as controls and two concentrations of Brassicapeptin A diluted in DMSO at 1 ng/mL and 1 μg/mL. Additionally, non-transfected protoplasts inoculated with water were used as controls. All protoplast conditions were incubated for 16 hours at 21 °C in the dark and harvested by centrifugation at 1000 x g. The supernatant was removed, and protoplasts were lysed by adding 250 μL of 2X lysis reagent (Promega, E1531). The LUC activity of samples was measured in a luminometer (Centro, LB960) for 1 second/sample using a 96-well plate in which 50 μL of protoplast lysate were mixed with 50 μL of the LUC substrate (Promega, E1501). For each condition, we 4 biological replicates and 4 technical replicates were included.

### Modified Burkholder assays
Modified Burkholder assays to determine the antagonistic potential of R401 and its mutants against 8 core microbiota members of *Arabidopsis* roots, including 6 bacteria and two fungi. The screen was carried out as described in Getzke et al. [24]. Strains were cultivated in 50% TSB until turbidity, stored at 4 °C and diluted 1:10 in 50% TSB one day before the experiment. Bacterial cultures were pelleted at 4000 rpm for 15 min. The resulting bacterial pellets were subsequently washed 3 times and resuspended in 1 ml 10 mM MgCl₂. OD₆₀₀ were measured and set depending on the strain. One hundred microliters bacterial culture were inoculated per 50 ml 25% TSA. After drying, up to nine different 3 μl droplets of bacterial suspensions with 0.4 OD₆₀₀ were applied with equal distances. For all experiments, plates were incubated at 25 °C for up to 96 hours and photographs were taken thereafter for quantitative image analysis. The size of the halo of inhibition was measured using ImageJ with up to five separate measurements, which were subsequently averaged to reduce variation. For *Fusarium oxysporum* F212 and *Macrophomina phaseolina* F80, pieces of 7-14 days old mycelium were transferred to pre-weighed sterile 2 mL screw cap tubes containing one and approx. 15 steel beads of 3 mm and 1 mm diameter, respectively. Per 50 mg harvested fungal mycelium, 1000 μL of sterile 10 mM MgCl₂ were added. The mycelium was subsequently grinded in a paint shaker at approx. 600 rpm for at least 10 min until homogeneous. The resulting slurry was used to inoculate 100% Potato Glucose Agar medium to a final concentration of 50 μg/ml.

### Minimum inhibitory concentration
Determination of the minimum inhibitory concentration (MIC) of brassicapeptin A was carried out by micro-broth-dilution assays in 96 well plates as previously reported[80]. Briefly, brassicapeptin A was dissolved in dimethyl sulfoxide (DMSO, Carl Roth GmbH + Co., Karlsruhe, Germany) with a concentration of 30 μM and tested in a triplicated 1:2-fold dilution series (64 to 0.03 μg/mL). Dilution series of rifampicin, tetracycline, and gentamicin (all Sigma -Aldrich, St. Louis, MO, USA) at the same concentrations were prepared as positive controls for *Escherichia coli* ATCC25922, *Staphylococcus aureus* ATCC25923 and *Listeria monocytogenes* DSM20600. Instead of gentamicin, isoniazid was used for *Mycobacterium smegmatis* ATCC607 assays at the same concentration range. *M. smegmatis* assays were incubated for 48 h (37 °C, 180 rpm, 85% rel. humidity r.H.) before cell viability was measured using BacTiter-Glo™ (BTG) according to the manufacturer's recommendations (Promega, Walldorf, Germany). For *Septoria tritici* MUCL45408, *Botrytis cinerea* HAG001286 and *Colletotrichum coccodes* DSM62126 tebuconazole (Cayman Chemical Company, Ann Arbor, MI, USA), amphotericin B (Sigma- Aldrich) and nystatin (Sigma Aldrich) were used as positive controls. For fungal assays, a previously prepared spore solution was diluted to $1 \times 10^5$ spores/mL in potato dextrose medium (Sigma-Aldrich). *Septoria* assay plates were incubated for 4 days, while *Botrytis* and *Colletotrichum* assays were only incubated for 48 h (all at 25 °C, 180 rpm and 85% r.H.) before cell viability was assessed by ATP quantification using BTG.

Additionally, we assessed the potency of brassicapeptin A against 8 microorganisms isolated from the rhizosphere of *A. thaliana*. Bacterial isolates were screened in tryptic soy broth using an assay inoculum of $5 \times 10^5$ cells/mL. Microtiter plates were incubated at 28 °C for 1-3 days and reference antibiotics were ceftazidime (LKT Laboratories, Inc., St. Paul, Minnesota), ciprofloxacin (Cayman Chemical Company) and gentamicin. Either cell viability (BTG) or turbidity measurements were used to determine growth inhibition. Fungal strains F212 and F80 were screened in potato glucose broth at 25 °C for 2 days (F212) or 3 days (F80). For strain F212 a spore solution was prepared and used for the assay inoculation as described above. The assay inoculum of strain F80 was prepared by diluting the homogenized pre-culture 1:4800. For both fungi the same reference antimycotics (tebuconazole, amphotericin B, nystatin) were used. Readout was done by cell viability (F212) or turbidity assessment (F80). The determined MICs for brassicapeptin A can be found in Supplementary Data 3.

### Statistical analyses
All statistical analyses were conducted in R 4.1.2. Data visualisation was conducted using the ggplot2 package (as part of the Tidyverse) or the ComplexHeatmap package. As nonparametric tests, Kruskal-Wallis followed by Dunn's post-hoc test and Benjamini-Hochberg (BH) adjustment for multiple comparisons from the PMCMRplus package (Pohlert, 2022) were used. The respective statistical tests are indicated in each figure description. Significance was indicated by significance group ($p \leq 0.05$). No statistical methods were used to pre-determine sample sizes. Halo size quantification in modified Burkholder experiments, and root length measurements were performed blinded using the Fiji package of ImageJ. Colony counts of R401 were performed blinded. RNA sequencing data processed as described above and further analysed and visualized as previously described[6]. GO Term enrichment was conducted as indicated in the respective figure description or results section. Figures were assembled in Adobe Illustrator.

### Reporting summary
Further information on research design is available in the Nature Portfolio Reporting Summary linked to this article.

## Data availability

RNAseq read data are available at GEO accession: GSE242479. All included data is also accessible from the link in the code availability section. A spreadsheet with all exact *p*-values is provided in the Source Data file. Source data are provided with this paper.

## Code availability

All codes and respective data generated for this study are available at https://github.com/scriptsFG/Getzke-Wang-et-al-2023.git.

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

## Acknowledgements

This work was supported by funds to S.H. from European Research Council starting and consolidator grants (MICRORULES 758003 and

MICROBIOSIS 101089198). It includes also funds to S.H. and P.S.-L. from the Max Planck Society, the Cluster of Excellence on Plant Sciences (CEPLAS) and the Priority Programme: Deconstruction and Reconstruction of the Plant Microbiota (SPP DECRyPT 2125; project P.S.-L.: SCHU 799/8-1; project S.H.: HA 8169/2-2), both funded by the Deutsche Forschungsgemeinschaft. Work in the Schäberle lab was supported by the German Federal Ministry of Education and Research (BMBF). L.W. was funded by the China Scholarship Council (CSC NO. 201908080177). We thank the Max Planck-Genome-Centre Cologne for advising and performing the RNA sequencing. We also thank Brigitte Pickel for her support in halo size quantifications and Flowpot harvest and Maria Patras for her support in Marfey´s analysis. Finally, thanks to Neysan Donnelly for editing this manuscript.

## Author contributions

F.G., S.H., T.F.S. and P.S.L. initiated the project. S.H. and T.F.S. supervised the project. F.G. and S.H. designed the experiments. L.W., N. B. and T.F.S. isolated and elucidated brassicapeptins. N.B. performed in vitro salt experiments with WT R401. F.G, N.D and G.C. performed Flowpot experiments and determined bacterial load on plant tissues. F.G. and H.W. performed CAS and the initial agar plate experiment. F.M. performed initial RNA sequencing data analysis including differential gene expression analysis. F.G. performed all further RNA sequencing analyses. F.G. and G.C. performed brassicapeptin A experiments. P.T.A. performed modified Burkholder assay screen. M.M. provided MIC data. F.G., L.W., G.C, P.T.A. and S.H. generated the figures. L.W. provided the original draft on brassicapeptin structure elucidation. F.G. and S.H. wrote the manuscript, with inputs from all co-authors.

## Funding

## Competing interests

The authors declare no competing interests.
