## [Peer Review File · Nature Communications]

Physiochemical interaction between osmotic stress and a bacterial exometabolite promotes plant diseaseReviewer #1 (Remarks to the Author):

I really enjoyed reading this manuscript reporting the shift from plant beneficial to pathogenic of *Pseudomonas brassicacearum* under salt/osmotic stress conditions. These interesting findings are supported by strong data and the study was very well designed. It provides new insights into the unsuspected behavioral instability of some key members of plant microbiomes. This work must certainly be considered for publication in *Nat Commun* but some points deserve further discussion.

I have first some specific remarks:

L152: I'm not sure that non-specialists will get easily the meaning of "Given the lack of genes encoding the type-III secretion system in the genome of R401 we hypothesized that this strain might use specialized metabolites as virulence factors." A few more words explaining that T3SS are used upon contact to deliver effectors should be added.

L171: "...sypC-dependent detrimental activity of R401 under salt stress was retained in plants co-cultured with a synthetic microbial community..." Yes but in that case, there is also loss of beneficial effect... Difficult to explain but should be mentioned I think.

L161: Did you verify by LC-MS the loss of brassicapeptin production in the Δ sypC mutant? And also in other mutants described later? I cannot retrieve this info also from the M&M.

L180: Contribution of sypC in the inhibition of some microbes: I assume it can be deduced (hardly) from the size of halos in Suppl fig 4b but then why the Δ sypC mutant seems not affected compared to the WT? Did you test pure brassicapeptin for antimicrobial activity against the same targets (as you purified the molecule in consistent amounts for structure determination)?

L200: "While colonization of R401 WT remained stable, ... the Δ sypC mutant showed impaired root colonization in salt-treated vs. control tomato plants" OK but I could not see data showing colonization of the Δ sypC mutant compared to WT on *Arabidopsis* under salt stress and in normal conditions? This is important in order to definitely rule out the hypothesis that loss of detrimental effect is due to much lower colonization and thus insufficient Δ sypC populations... I understand that according to Fig3d, growth of both strains do not really suffer from high NaCl in liquid medium but the situation may be different on roots because salt stress may impact indirectly by triggering some response in the host plant which in turn would restrict root invasion by Δ sypC. I agree it is not that obvious but cannot be fully rejected...

L210: Antismash is quite powerful and can to some extent predict the amino acid sequence in the peptide part according to the adenylation domains and the lipopeptides of the peptin and factin families are synthesized co-linearly with the order of the modules present in their respective NRPS enzymes. So did you get some prediction of the structure just based on genome mining? See [10.1080/1040841X.2020.1794790](https://doi.org/10.1080/1040841X.2020.1794790) and [10.1128/msystems.00988-22](https://doi.org/10.1128/msystems.00988-22) for recent comprehensive reviews on *Pseudomonas* lipopeptides.

Fig 3a: as in *P. syringae*, both BGCs are tightly clustered, why not mentioning the annotation brassicamycin? Just to be complete...

L239 and discussion: "brassicapeptins represent a novel subgroup" OK but it is more similar to cicho-peptin (or corpeptin) from *P. cichorii*, another phytopathogen belonging to *P. syringae* complex, which are also made of 22 AAs but differ from SP22 by the size of its macrocycle (five versus eight residues).

L257: syringomycin as "structurally-related" lipopeptide. It depends the point of view; OK they are both lipopeptides but syringomycin is 9 AA and fully cyclized peptide moiety, so quite different! Syringopeptin SP22 much more close...

L259: Ref 48: I have noting against this paper but quite old, more recent works can be used I guess

L259-61: "brassicapeptin A-induced disruption of ion homeostasis, combined with increased

osmotic pressure in the root environment, likely contribute to R401-induced disease symptoms in salt-stressed plants. I understand but this remains quite hypothetical and "likely" sounds too strong.

The ability to co-produce two (or three) different lipopeptides is very common if not a trademark of phytopathogenic species in *Pseudomonas*. Consistently, these strains co-produce a second lipopeptide from the mycin type. This could be added somewhere.

General comments:

The conclusion of the work can be as expressed L311-13: "Collectively, this indicates that the direct interaction of a bacterial specialized metabolite and high osmotic concentrations in the environment is sufficient to explain R401 detrimental activity on salt-sensitive plants"

I have nothing against this statement but no real explanation of how it works is provided except in L262-4: "... hypothesis that brassicaeptin A-induced disruption of ion homeostasis at the root interface enhances salt stress and promotes bacterial colonization, thereby leading to disease".

I think there is not enough data to support this since leakage on roots has not been performed (and the situation can be very different from leaves because membrane lipids differ). In line with the data provided here, the CLPs produced by pathogenic *P. syringae*, *P. fuscovaginae*, *P. corrugata*, and *P. cichorii*, contribute to virulence via their phytotoxic activities due to the amphipathic structure of peptin-type lipopeptides. Like for brassicaeptin, it allows insertion into the lipid bilayers of membranes to form pores, which induces electrolyte leakage but also subsequent death of plant cells. Leakage OK but did the authors test the effect of brassicaeptin treatment on plant cell viability in addition to leakage?

Some colonization data are missing (see above). Not clear what means "enhances salt stress" and what is the link with "promotes bacterial colonization"?

Is brassicaeptin produced in consistent amounts by *Pseudomonas* colonizing roots? And what are the amounts secreted by the bacterium in high salinity or osmotic stress conditions? Can we imagine that in such conditions, *sypC* is much more expressed and thus, that the higher toxicity of the bacterium is just due to higher amounts of the lipopeptide creating more pores in root cell membranes? Just in a process not directly dependent of salt concentration/osmotic stress... The fact that "purified brassicaeptin A was sufficient to negatively impact shoot and root phenotypes already in the absence of salt stress" is in support to this assumption...

In other words, shift to a pathogenic lifestyle may be only related to the amounts of CLP produced by the colonizing strain.

Mycins and peptins can work in synergy and altering the production of one or the other may induce a strong reduction of virulence, did the authors test or have indications on such possible synergistic effect of Brassicamycin ?

Reviewer #2 (Remarks to the Author):

The topic of this manuscript is the interplay between abiotic stress conditions and disease development in plants. Specifically, previous research identified a root-associated pseudomonad, R401, that causes disease only in specific abiotic conditions (in MS agar but not in agricultural soil). The authors demonstrated that salt stress in the MS agar affected the disease appearance in the presence of R401, and a pore-forming toxin related to syringomycin in R401 is responsible for the condition-specific disease phenotype. The manuscript concludes that the toxin reduces the capacity of the plant to regulate ion homeostasis in the high salt environment.

This manuscript provides a clear mechanistic example of how an abiotic condition can influence the interaction between a plant and a microbe. I really liked the paper, and it has changed how I view

the effects of pore-forming toxins in pathogenicity. In retrospect, however, I wonder if the finding that the pores formed by syp-syr make plants vulnerable to ionic stress is the a priori expectation. Indeed, a substantial literature on syringomycin-syringopeptin firmly shows that these toxins disrupt eukaryotic membranes. On a related note, there is of course a large body of literature demonstrating that plant-associated pseudomonads differ substantially in their toxin profiles and encode suites of different pathogenicity-associated toxins (Melnyk et al. 2019 for example). This paper did not demonstrate in my view that the mechanism they identify is particularly pervasive instead of one of many mechanisms underlying pathogenicity and the disease triangle. While I quite like the example detailed in this paper, it's not clear to me that this manuscript substantially advances our understanding of mechanisms underlying the disease triangle.

Major comment:

- The manuscript provides mixed evidence that sypC improves the fitness of the bacteria in a salt-stressed environment. The results seem inconsistent – the gene improves fitness in one plant and not another. Please clarify in the text (i) under which conditions the toxin improves the bacterial fitness and include (ii) discussion of why these results are not transferrable across hosts.
- The manuscript text largely excludes p-values and mentions of statistical tests. These details are relegated to the legends. Throughout the manuscript I found myself wanting to see the statistical robustness of a conclusion presented in the main text.

Minor comments:

Line 121: Should read "Although this activation is associated..."

Virulence: (line 134) What is your definition of virulence?

Line 137: Should read "was recently shown to fully depend"

Line 169: Please provide p-value and statistical test after result is described.

Line 183: The authors argue that the acquisition of syp-syr will affect plant disease emergence in complex soil systems.

Line 234: Correct language of "the here discovered"

Line 258: should read "likely functions"

Line 261: Should read "Likely contributing to..."

Figure 2b: These networks are hard to interpret. It's not clear to me what the major message is. I recommend simplifying to distill to major points.

Figure 3a: The gene plots are quite simplified. It's not clear that there is homology in between R401 and B728a given that nearly all genes are grayed out without names.

Reviewer #1 (Remarks to the Author):

I really enjoyed reading this manuscript reporting the shift from plant beneficial to pathogenic of *Pseudomonas brassicacearum* under salt/osmotic stress conditions. These interesting findings are supported by strong data and the study was very well designed. It provides new insights into the unsuspected behavioral instability of some key members of plant microbiomes. This work must certainly be considered for publication in *Nat Commun* but some points deserve further discussion.

We thank the reviewer for the positive and very constructive feedback. We have very carefully addressed the points of concerns by performing several new experiments and analyses. In short:

- **We showed that *brpC* (previously referred to as *sypC*) is a root colonization determinant (in *A. thaliana*).**
- **We performed *in vitro* experiment with R401WT grown in liquid medium supplemented with salt and showed that brassicaeptin production is salt-inducible.**
- **We performed new halo of inhibition assays, together with brassicaeptin A MIC assays and demonstrate that brassicaeptin A has moderate antimicrobial activity and inhibits the growth of several environmental pathogens (bacterial and fungal) and some root microbiota members.**
- **We validated that brassicaeptin A compromises *A. thaliana* cell viability using a protoplast transfection assay with a luciferase construct.**
- **We confirmed that *brpC* contributes to R401 detrimental activity in the agar-based gnotobiotic plant system.**

I have first some specific remarks:

L152: I'm not sure that non-specialists will get easily the meaning of "Given the lack of genes encoding the type-III secretion system in the genome of R401 we hypothesized that this strain might use specialized metabolites as virulence factors." A few more words explaining that T3SS are used upon contact to deliver effectors should be added.

We agree, we have modified the sentence: "Inspection of the genome of R401 revealed the absence of genes encoding the type-III secretion system, indicating that intracellular delivery of bacterial effectors via this machinery is likely not the cause of R401 detrimental activity in salt-treated plants"(see lines 161 – 165)

L171: "...*sypC*-dependent detrimental activity of R401 under salt stress was retained in plants co-cultured with a synthetic microbial community..." Yes but in that case, there is also loss of beneficial effect... Difficult to explain but should be mentioned I think.

We agree that the plant growth-promoting (PGP) effect of the $\Delta brpC$ (previously referred to as *sypC*) was no longer observed in a SynCom context. This is likely explained by the fact that the SynCom also had PGP activity under salt stress, which likely masked the PGP effect of $\Delta brpC$ due to functional redundancy in a community context.

L161: Did you verify by LC-MS the loss of brassicaeptin production in the $\Delta sypC$ mutant? And also in other mutants described later? I cannot retrieve this info also from the M&M.

Thanks for pointing this out. Abolition of brassicaeptin production in the $\Delta brpC$ mutant was corroborated by UPLC-HRMS. In the new version, the info can be found in the results section, in Fig. 3b and a corresponding methods paragraph was added. Fig. 3b was modified to show chromatogram traces of the R401 WT and $\Delta brpC$ mutant. (L174-175, L547-550).

L180: Contribution of *sypC* in the inhibition of some microbes: I assume it can be deduced (hardly) from the size of halos in Suppl fig 4b but then why the $\Delta sypC$ mutant seems not affected compared to the WT? Did you test pure brassicaeptin for antimicrobial activity against the same targets (as you purified the molecule in consistent amounts for structure determination)?

We agree with the reviewer. We have now performed two new important experiments.

- On one hand, we have performed a more comprehensive halo of inhibition screen using four R401 strains (WT, Δ brpC, Δ pvdY Δ phlD, Δ pvdY Δ phlD Δ brpC) and more target strains that now include 6 bacterial and 2 fungal strains from the core root microbiota of *A. thaliana*. Notably, we used a R401 triple mutant impaired in the production of pyoverdine, DAPG, and brassicapeptin (Δ pvdY Δ phlD Δ brpC vs Δ pvdY Δ phlD Fig. 6b) in order to abolish the dominant inhibitory activity of DAPG and pyoverdine, which co-explained >70% of R401 antagonistic activity based on previous measurements of R401 inhibitory halos (Getzke et al. PNAS 2023). We made it clearer in the text (See lines 297 - 300). We think that comparing WT and Δ brpC only is not informative because DAPG (and to a lesser extent pyoverdine) is masking the weaker inhibitory effect of brassicapeptin.

-On another hand, we also conducted MIC determinations of brassicapeptin against the exact same microbial isolates (n = 8) as well as some environmental pathogens (n = 7) using micro-broth-dilution assays. The screening protocol e.g. growth medium, incubation time and temperature as well as the read-out method, were optimized for each of the test strains. We included the data table in the manuscript (Fig. 6c and Supplementary table 3). Our results unambiguously demonstrate that brassicapeptin has moderate, yet multi-kingdom antimicrobial activity. We think this is an important conclusion and we therefore decided to include these data in the main text together with a new main figure (See lines 293 – 322, Fig. 6c, Supplementary Table 3).

L200: "While colonization of R401 WT remained stable, ... the Δ sypc mutant showed impaired root colonization in salt-treated vs. control tomato plants" OK but I could not see data showing colonization of the Δ sypc mutant compared to WT on Arabidopsis under salt stress and in normal conditions? This is important in order to definitely rule out the hypothesis that loss of detrimental effect is due to much lower colonization and thus insufficient Δ sypc populations... I understand that according to Fig3d, growth of both strains do not really suffer from high NaCl in liquid medium but the situation may be different on roots because salt stress may impact indirectly by triggering some response in the host plant which in turn would restrict root invasion by Δ sypc. I agree it is not that obvious but cannot be fully rejected...

The reviewer is correct and this information was initially not included in the earlier version because this analysis was only performed for Micro-Tom and Lotus japonicus. Therefore, we have performed a new *A. thaliana* recolonization experiment with Δ brpC and WT strain under +/- 100mM salt. We actually observed that in *A. thaliana* roots, bacterial load is reduced in the Δ brpC mutant (vs WT) under both + and - 100mM salt. This indicates that brpC is a root colonization determinant and that brassicapeptin production is needed for efficient root colonization. Our results indicate that the lack of disease in Δ brpC could be explained by the reduced bacterial load in roots but clearly this difference in bacterial load cannot be the sole reason explaining abolishment of disease. Actually, we always observed multiple times independently (See Supplementary Fig. 1c, Fig. 3b, Supplementary Fig. 6c,d) that bacterial load of the WT strain does not differ between 0 mM salt (no disease observed) and 100 mM salt (disease observed) in planta. We also observed similar results in vitro (see Fig. 3f). Therefore, although disease symptoms were observed in salt-stressed plants, this disease phenotype was NOT associated with bacterial overgrowth compared to plants grown under 0 mM salt.

L210: Antismash is quite powerful and can to some extent predict the amino acid sequence in the peptide part according to the adenylation domains and the lipopeptides of the peptin and factin families are synthesized co-linearly with the order of the modules present in their respective NRPS enzymes. So did you get some prediction of the structure just based on genome mining? See 10.1080/1040841X.2020.1794790 and 10.1128/msystems.00988-22 for recent comprehensive reviews on Pseudomonas lipopeptides.

Fig 3a: as in *P. syringae*, both BGCs are tightly clustered, why not mentioning the annotation brassicamycin? Just to be complete...

Indeed, the predictions by antiSMASH already gave a valid prediction for most of the amino acids. We now added a supplementary table (i.e., Supplementary Table 2), comparing the in silico prediction of the A domains specificities of the brassicapeptin BGC to the observed incorporated amino acids. The prediction of the putative compound brassicamycin is given as well; however, the compound was not detected yet. Fig. 3 was modified accordingly to indicate the tightly clustered –peptin and –mycin parts of the Syringo- and Brassica- BGCs. However, it

has to be kept in mind that the BGC is annotated together (MIBiG entry BGC0000437, derived from NCBI GenBank CP000075.1) and spans the genes involved in biosynthesis of syringopeptin and syringomycin as well as corresponding regulatory genes. Therefore, we decided to highlight the core NRPS genes of both compounds. The same was done for the here described brassicaeptin and putative –mycin. Additionally, we created a table in which the genes are mentioned and the predicted function in the biosynthesis of the metabolites is given. This is also compared between the Syringopeptin and Brassicaeptin BGCs.

L239 and discussion: “brassicaeptins represent a novel subgroup” OK but it is more similar to cichozeptin (or corzeptin) from *P. cichorii*, another phytopathogen belonging to *P. syringae* complex, which are also made of 22 AAs but differ from SP22 by the size of its macrocycle (five versus eight residues).

We agree with the reviewer that the similarity is higher to cichozeptin. We first named the probably best-known example syringopeptin. In the new version, it is rephrased towards « ... represent a novel sub-group of cyclic syringopeptin-type lipopeptides. » Then, the similarity to cichozeptin and corzeptin is explained in more detail. Furthermore, the cichozeptin A and B, as well as the corzeptin A and B structures were incorporated into supplementary Fig. 8, to enable quick comparison of these peptides. (see also lines 257-260)

L257: syringomycin as “structurally-related” lipopeptide. It depends the point of view; OK they are both lipopeptides but syringomycin is 9 AA and fully cyclized peptide moiety, so quite different! Syringopeptin SP22 much more close...

We thank the reviewer for this point. This was an error, of course the bigger -peptin structures should be compared to each other and not –peptins with -mycins. In the new version (see point before) this is rephrased.

L259: Ref 48: I have noting against this paper but quite old, more recent works can be used I guess

Thanks, we checked this carefully but we still believe that this review article (to our knowledge) comprehensively describe the mode of action of syringopeptin. We have also cited Carpaneto A, et al. J Membrane Biol 188, 237-248 (2002) and Agner G, et al. Bioelectrochemistry 52, 161-167 (2000). If the reviewer has another manuscript in mind, we will be happy to include it in the next version.

L259-61: “brassicaeptin A-induced disruption of ion homeostasis, combined with increased osmotic pressure in the root environment, likely contribute to R401-induced disease symptoms in salt-stressed plants. I understand but this remains quite hypothetical and “likely” sounds too strong.

We have performed a new experiment using protoplast transfection assay with a luciferase construct that we used as proxy to measure brassicaeptin-induced cell death (see below). We are now confident that this statement is not overstated based on earlier work performed on similar cyclic lipopeptides, based on the fact that brassicaeptin A promotes electrolyte leakage from leaf discs and also compromised leaf protoplast viability (see below, See also Fig. 5e, lines 282 – 283, and lines 647 – 664).

The ability to co-produce two (or three) different lipopeptides is very common if not a trademark of phytopathogenic species in *Pseudomonas*. Consistently, these strains co-produce a second lipopeptide from the mycin type. This could be added somewhere.

We agree with the reviewer and indeed anticipated the production of a second lipopeptide from the mycin type. However, we did not detect the corresponding molecule yet. A sentence referring to the co-occurrence of peptin and mycin type molecules was now added. (lines 168 - 169).

General comments:

The conclusion of the work can be as expressed L311-13: “Collectively, this indicates that the direct interaction of a bacterial specialized metabolite and high osmotic concentrations in the environment is

sufficient to explain R401 detrimental activity on salt-sensitive plants”

I have nothing against this statement but no real explanation of how it works is provided except in L262-4: “... hypothesis that brassicapeptin A-induced disruption of ion homeostasis at the root interface enhances salt stress and promotes bacterial colonization, thereby leading to disease”.

We now make this point very clear across the manuscript (see lines 81 – 85, lines 197 – 200, lines 288 – 291, lines 319 – 322, lines 374 - 385). Our results indicate that brpC is needed for bacterial colonization and disease. All our data point to the conclusion that brassicapeptin production at roots is needed for bacterial proliferation but the disease symptoms are more likely to be a direct consequence of brassicapeptin-induced ion disruption and hyper susceptibility to salt rather than bacterial overgrowth. We also obtained new in vitro data indicating that brassicapeptin production is salt-inducible (see below, see also Supplementary Fig. 4). All together, these data indicate that plants colonized by R401 become supersusceptible to salt stress due to brassicapeptin-induced disruption of ion homeostasis. This is fully consistent with the additive effect of salt stress and purified brassicapeptin A, leading to dead plants in agar plate assays.

I think there is not enough data to support this since leakage on roots has not been performed (and the situation can be very different from leaves because membrane lipids differ). In line with the data provided here, the CLPs produced by pathogenic *P. syringae*, *P. fuscovaginae*, *P. corrugata*, and *P. cichorii*, contribute to virulence via their phytotoxic activities due to the amphipathic structure of peptin-type lipopeptides. Like for brassicapeptin, it allows insertion into the lipid bilayers of membranes to form pores, which induces electrolyte leakage but also subsequent death of plant cells. Leakage OK but did the authors test the effect of brassicapeptin treatment on plant cell viability in addition to leakage?

We have performed a new experiment based on “protoplast transfection assay with a luciferase construct” that we used as proxy to measure brassicapeptin-induced cell death (see below and Fig. 5e). This new experiment indicates that brassicapeptin (1µg/mL) was sufficient to fully abolish luciferase activity. This method has been used multiple times and represents a good proxy to assess cell viability (Saur et al. , 2019 Plant Methods 15:118). Note that we tried to perform ion leakage experiments in roots and also to generate protoplasts from roots. This was unfortunately not successful. Therefore, we used leaf discs and leaf-derived protoplasts to test for molecular function of brassicapeptin. We understand that this is rather indirect evidence of how brassicapeptin might function in roots. We have added a sentence in the discussion to make the reader aware of this limitation (see lines 369 – 370).

Some colonization data are missing (see above). Not clear what means “enhances salt stress” and what is the link with “promotes bacterial colonization”?

This experiment has been done (see above). It revealed that brpC is a root colonization determinant under both 0 mM NaCl and 100 mM NaCl in A. thaliana.

Is brassicapeptin produced in consistent amounts by *Pseudomonas* colonizing roots? And what are the amounts secreted by the bacterium in high salinity or osmotic stress conditions? Can we imagine that in such conditions, *sypC* is much more expressed and thus, that the higher toxicity of the bacterium is just due to higher amounts of the lipopeptide creating more pores in root cell membranes? Just in a process not directly dependent of salt concentration/osmotic stress... The fact that “purified brassicapeptin A was sufficient to negatively impact shoot and root phenotypes already in the absence of salt stress” is in support to this assumption...

In other words, shift to a pathogenic lifestyle may be only related to the amounts of CLP produced by the colonizing strain.

We really thank the reviewer for raising this important point. We could not quantify brassicapeptin production in planta between 0 mM NaCl and 100 mM NaCl. However, we performed a new in vitro experiment to test whether brassicapeptin production is salt-inducible. Notably, we observed that increased concentration of salt in the bacterial growth medium triggers brassicapeptin accumulation in the medium. Our results indicate that brassicapeptin production is therefore salt-inducible. Therefore, it is fully plausible that

disease symptoms observed under salt are directly linked to higher brassicapeptin amounts rather than higher bacterial load. We make it clear in the text as well (see lines X, Y, Z). However, our experiment using purified brassicapeptin A and also salt clearly indicate that the same concentration of brassicapeptin used for NaCl-treated and control plants leads to different phenotypic differences, clearly indicating that brassicapeptin production is more damageable for plants under salt. Therefore both higher brassicapeptin amount and salt are likely driving disease.

Mycins and peptins can work in synergy and altering the production of one or the other may induce a strong reduction of virulence, did the authors test or have indications on such possible synergistic effect of Brassicamycin ?

Despite great efforts, brassicamycin could not be detected and characterized. Therefore, we cannot provide data concerning the putative synergistic effects. Paragraphs were modified accordingly to address this detail (see lines 355 - 356). Our new data indicating that brpC is only partially contributing to R401 detrimental activity in the agar system suggesting that other pathogenicity determinant act additively to brpC to confer disease (see new Supplementary Fig. 10 and lines 264 – 268). Brassicamycin could be the missing link. We are generating now a R401 mutant lacking the brassicamycin core NRPS gene, as well as corresponding double mutants but it will take months until we have these results.

Reviewer #2 (Remarks to the Author):

The topic of this manuscript is the interplay between abiotic stress conditions and disease development in plants. Specifically, previous research identified a root-associated pseudomonad, R401, that causes disease only in specific abiotic conditions (in MS agar but not in agricultural soil). The authors demonstrated that salt stress in the MS agar affected the disease appearance in the presence of R401, and a pore-forming toxin related to syringomycin in R401 is responsible for the condition-specific disease phenotype. The manuscript concludes that the toxin reduces the capacity of the plant to regulate ion homeostasis in the high salt environment.

This manuscript provides a clear mechanistic example of how an abiotic condition can influence the interaction between a plant and a microbe. I really liked the paper, and it has changed how I view the effects of pore-forming toxins in pathogenicity.

We thank the reviewer for the positive comment. We have performed several new experiments that strengthen our earlier conclusions. In short:

- We performed in vitro experiment with R401WT grown in liquid medium supplemented with salt and showed that brassicapeptin production is salt-inducible.**
- We performed new halo of inhibition assays, together with brassicapeptin A MIC assays and demonstrate that brassicapeptin A has moderate antimicrobial activity and inhibits the growth of several environmental pathogens (bacterial and fungal) and some root microbiota members.**
- We validated that brassicapeptin A compromises *A. thaliana* cell viability using a protoplast transfection assay with a luciferase construct.**
- We confirmed that brpC contributes to R401 detrimental activity in the agar-based gnotobiotic plant system.**

In retrospect, however, I wonder if the finding that the pores formed by syp-syr make plants vulnerable to ionic stress is the a priori expectation. Indeed, a substantial literature on syringomycin-syringopeptin firmly shows that these toxins disrupt eukaryotic membranes.

We thank the reviewer for raising this point. We do believe that our experiment in agar-based gnotobiotic system using salt and purified brassicapeptin A is the best validation of our claim. Based on this experiment, we demonstrated that brassicapeptin A promotes osmotic stress, thereby leading to dead plants (See Fig. 5a-c).

As the reviewer mentioned, our results also provide several lines of evidence showing that brassicapeptin promotes electrolyte leakage and impede protoplast viability (see new protoplast transfection assay in Fig 5d). We do not think that these two conclusions are mutually exclusive. Indeed, brassicapeptin-induced disruption of ion homeostasis could be

highly detrimental under salt stress due to plant's inability to regulate salt toxicity. Note that we also provide new evidence that brassicaeptin is salt-inducible (at least in vitro, see below) and therefore, it remains also possible that the amount of brassicaeptin produced in roots is greater under salt stress, thereby promoting more cell damage and disease symptoms. We make it clear in the text (see lines 193 – 200).

On a related note, there is of course a large body of literature demonstrating that plant-associated pseudomonads differ substantially in their toxin profiles and encode suites of different pathogenicity-associated toxins (Melnyk et al. 2019 for example). This paper did not demonstrate in my view that the mechanism they identify is particularly pervasive instead of one of many mechanisms underlying pathogenicity and the disease triangle.

We fully agree that we could not assess the pervasiveness of this mechanisms across multiple Pseudomonas isolates. However, we provide a bioinformatic survey of the prevalence of the syp-syr operon in >1,000 Pseudomonas genomes and observed that the prevalence of this BGC is actually rare in Pseudomonas isolates. We also noted that some strains that encode this BGC show very similar phenotypes that our strain, such as the N2C3 that encodes this BGC and showed high pathogenicity in agar-based gnotobiotic system but not in a soil-based system (Wang et al. 2022). If the reviewer insists, we could test the detrimental effect of the N2C3 strain on soil-grown A. thaliana facing salt stress. However, this will not change our major conclusions.

While I quite like the example detailed in this paper, it's not clear to me that this manuscript substantially advances our understanding of mechanisms underlying the disease triangle.

We now provide a new in vitro experiment showing that brassicaeptin production is salt-inducible (see new Supplementary Fig. 4 and lines 196 – 200). Therefore salt-dependent regulation of brassicaeptin production in Pseudomonas may have an interplay with the salt adaptation strategy of the plant. This may also explain differences in susceptibility to this opportunistic plant pathogen throughout the different host plants. This also indicates that salt concentration and brassicaeptin production are likely interconnected, which potentially brings novel insights into the disease triangle concept.

Major comment:

- The manuscript provides mixed evidence that sypC improves the fitness of the bacteria in a salt-stressed environment. The results seem inconsistent – the gene improves fitness in one plant and not another. Please clarify in the text (i) under which conditions the toxin improves the bacterial fitness and include (ii) discussion of why these results are not transferrable across hosts.

Thanks for pointing this out. This information was initially not included (for A. thaliana) in the earlier version because this analysis was only performed for Micro-Tom and Lotus japonicus. Therefore, we have performed a new A. thaliana recolonization experiment with $\Delta brpC$ and WT strain under +/- 100mM salt. The results are shown in Fig. 3e. We actually observed that in A. thaliana roots, bacterial load is reduced in the $\Delta brpC$ mutant (vs WT) under both + and - 100mM salt. This was also the case in tomato (although to a lesser extend under - 100 mM NaCl). However, brpc was clearly not important for root colonization of L. japonicus. We make it clear in the text now (see lines 213 – 215 and lines 220 – 223)

This indicates that brpC is a root colonization determinant and that brassicaeptin production is needed for efficient root colonization at least in A. thaliana and tomato.

In the context of the WT strain, we always observed multiple times independently (See Supplementary Fig. 1c, Fig. 3b, Supplementary Fig. 6c,d) that bacterial load of the WT strain does not differ between 0 mM salt (no disease observed) and 100 mM salt (disease observed) in planta. We also observed similar results in vitro (see Fig. 3f). Therefore, although disease symptoms were observed in salt-stressed plants, this disease phenotype was NOT associated with bacterial overgrowth compared to plants grown under 0 mM salt.

- The manuscript text largely excludes p-values and mentions of statistical tests. These details are relegated to the legends. Throughout the manuscript I found myself wanting to see the statistical

robustness of a conclusion presented in the main text.

The p- values are now indicated in the text. The test used is always described in the figure legend due to length constraints imposed by Nature Comms.

Minor comments:

Line 121: Should read “Although this activation is associated...”

Done

Virulence: (line 134) What is your definition of virulence?

We agree that based on our conclusions, the term virulence is not the most adapted term. We now used disease or infection instead. We also make it clear that disease (in our specific case), is referred to as the combined detrimental effect of a biotic (bacterial-derived metabolite) and an abiotic (NaCl) factor.

Line 137: Should read “was recently shown to fully depend”

Done

Line 169: Please provide p-value and statistical test after result is described.

Done. The test used is always described in the figure legend due to length constraints imposed by Nature Comms.

Line 183: The authors argue that the acquisition of syp-syr will affect plant disease emergence in complex soil systems.

We are not sure about the concern here. We showed that brpC is 1) a root colonization determinant, 2) a disease determinant under salt stress in soil systems, 3) dominantly functioning irrespective of the presence of microbial competitors 4) contributing to the inhibitory activity of R401. Therefore, we delineate how a single bacterial molecule can have multiple independent effects on organisms that evolved in at least three kingdoms of life (plants, bacteria, fungi), thereby contributing to bacterial competitiveness at roots and promoting plant susceptibility to osmotic stress. Based on this, we think that this conclusion is not overstated (see also lines 429 – 435).

Line 234: Correct language of “the here discovered”

Done

Line 258: should read “likely functions”d

Done

Line 261: Should read “Likely contributing to...”

Done

Figure 2b: These networks are hard to interpret. It's not clear to me what the major message is. I recommend simplifying to distill to major points.

We have moved this figure panel to supplementary information and present a much-simplified version that illustrate our main point: down-regulation of immune process in salt-treated plants is shoot-specific.

Figure 3a: The gene plots are quite simplified. It's not clear that there is homology in between R401 and B728a given that nearly all genes are grayed out without names.

We fully agree. We present an updated version of the figure (see Fig. 3a) that now highlighted core NRPS genes involved in the biosynthesis of syringopeptin/brassicapeptin and syringomycin/brassicamycin.

Reviewer #1 (Remarks to the Author):

The authors considered very carefully all the concerns and performed the necessary additional experiments to provide data in support to all statements. I congratulate the authors for this very interesting study, which is now, in my opinion, acceptable for publication in Nature Communications.

Reviewer #2 (Remarks to the Author):

The authors have thoroughly responded to the suggestions and questions from the first round of reviews. The revised manuscript is an excellent demonstration of a mechanism of interaction between the microbe and environment to cause disease.

Reviewer #1 (Remarks to the Author):

The authors considered very carefully all the concerns and performed the necessary additional experiments to provide data in support to all statements. I congratulate the authors for this very interesting study, which is now, in my opinion, acceptable for publication in Nature Communications.

Many thanks

Reviewer #2 (Remarks to the Author):

The authors have thoroughly responded to the suggestions and questions from the first round of reviews. The revised manuscript is an excellent demonstration of a mechanism of interaction between the microbe and environment to cause disease.

Many thanks